# An oviduct-on-a-chip provides an enhanced in vitro environment for zygote genome reprogramming

Marcia A.M.M. Ferraz [1,2], Hoon Suk Rho [3], Daiane Hemerich[4,5], Heiko H.W. Henning[6], Helena T.A. van Tol[1], Michael Hölker[7,8], Urban Besenfelder[9], Michal Mokry[10], Peter L.A.M. Vos[1], Tom A.E. Stout [6], Séverine Le Gac[3] & Bart M. Gadella [1,2]

Worldwide over 5 million children have been conceived using assisted reproductive technology, and research has concentrated on increasing the likelihood of ongoing pregnancy. However, studies using animal models have indicated undesirable effects of in vitro embryo culture on offspring development and health. In vivo, the oviduct hosts a period in which the early embryo undergoes complete reprogramming of its (epi)genome in preparation for the reacquisition of (epi)genetic marks. We designed an oviduct-on-a-chip platform to better investigate the mechanisms related to (epi)genetic reprogramming and the degree to which they differ between in vitro and in vivo embryos. The device supports more physiological (in vivo-like) zygote genetic reprogramming than conventional IVF. This approach will be instrumental in identifying and investigating factors critical to fertilization and pre-implantation development, which could improve the quality and (epi)genetic integrity of IVF zygotes with likely relevance for early embryonic and later fetal development.

[1] Department of Farm Animal Health, Faculty of Veterinary Medicine, Utrecht University, Yalelaan 104, 3584 CM Utrecht, The Netherlands. [2] Department of Biochemistry and Cell Biology, Faculty of Veterinary Medicine, Utrecht University, Yalelaan 79, 3584 CM Utrecht, The Netherlands. [3] Applied Microfluidics for Bioengineering Research, MESA+ Institute for Nanotechnology and MIRA Institute for Biomedical Technology and Technical Medicine, University of Twente, Enschede 7500 AE, The Netherlands. [4] Division Heart and Lungs, Department of Cardiology, University Medical Center Utrecht, Heidelberglaan 100, 3584 CX Utrecht, The Netherlands. [5] CAPES Foundation, Ministry of Education of Brazil, Brasília, DF 70040-020, Brazil. [6] Department of Equine Sciences, Faculty of Veterinary Medicine, Utrecht University, Yalelaan 112, 3584 CM Utrecht, The Netherlands. [7] Research Station Frankenforst, Faculty of Agriculture, University of Bonn, Versuchsgut Frankenforst 4, 53639 Koenigswinter, Germany. [8] Department of Animal Breeding and Husbandry, Institute of Animal Science, University of Bonn, Endenicher Allee 15, 253115 Bonn, Germany. [9] Institute of Animal Breeding and Genetics, University of Veterinary Medicine Vienna, 1210 Vienna, Austria. [10] Epigenomics Facility, University Medical Center Utrecht, Heidelberglaan 100, 3584 CX Utrecht, Netherlands. Correspondence and requests for materials should be addressed to B.M.G. (email: b.m.gadella@uu.nl)

I n vitro embryo production (IVP) in mammals involves a marked change in the microenvironment to which the early embryo is exposed and, despite considerable improvements in the success of assisted reproductive technologies (ART), IVP systems are still far from physiological. That these conditions are suboptimal is evidenced by substantial differences between embryo production in vitro and in vivo; depending on species, the former is associated with lower blastocyst per oocyte yields, reduced developmental competence, altered gene expression patterns, changes in epigenetic reprogramming and a reduced likelihood of successful pregnancy[1–10]. In vivo, the oviduct hosts a period in which the early embryo undergoes a reprogramming of its (epi)genome in preparation for the reacquisition of epigenetic marks in specific cell populations as they progress through differentiation[2,11,12]. This period of epigenetic reprogramming has proven to be extremely sensitive to changes in environmental conditions, such as compromised maternal health or an unhealthy diet[13]. Epigenetic reprogramming can also be disturbed by the conditions imposed by IVP, such as culture medium components, light, temperature and oxygen tension[2,4,5,14]. Many of the epigenetic effects associated with in vitro embryo production can be detected during the pre-implantation period[4,15,16]. Other effects only become apparent during later fetal or even post-natal development, and these include unbalanced fetal-placental development, abnormal fetal growth and abnormal metabolic responses or predilection to 'lifestyle' related diseases in neonatal or adult life[1,2,17,18].

Improvements in IVP, not only in terms of numbers of embryos produced, time to pregnancy and likelihood of live birth, but more specifically in terms of embryo quality and 'normality' are essential to safeguard the health of future generations of in vitro fertilization (IVF) offspring. In this light, it is somewhat surprising that the influence of the oviduct on mammalian embryo development has not been thoroughly investigated to inform the refinement of ART procedures[3]. We have hypothesized that, by mimicking an oviductal environment in vitro, the processes of fertilization and early embryo development would more closely resemble the physiological situation. A first attempt to this end was to create a three-dimensional (3D)-printed oviduct-on-a-chip culture chamber[19], which indeed showed that this approach can be used to optimize exclusive monospermic IVF, which is useful for improving IVP. However, next to this we discovered that routine materials used for 3D printing of chambers used in cell culture released toxic components (phthalates and ethylene-glycols) that arrested early embryo development of fertilized oocytes[20] while polydimethylsiloxane (PDMS) was not toxic. Therefore, we designed a microfluidic 'oviduct-on-a-chip platform' in which oviductal epithelial cells were cultured and maintains the morphological and functional structure, similar to the in vivo oviduct. The oviduct-on-a-chip also permits the production of bovine zygotes with a transcriptome and global methylation pattern resembling in vivo produced zygotes but dissimilar to conventional IVP zygotes.

## Results

### Oviduct-on-a-chip design.
Bovine oviduct epithelial cells (BOECs) rapidly lose their polarization and differentiation in 2D static culture[21–23]. To maintain in vivo-like morphology (a cuboidal to columnar pseudostratified epithelium with ciliated and secretory cells[24–26]) and function, alternative 3D culture methods have been described, e.g., using air–liquid interfaces[27–29], organoids[30], suspensions[24], and perfusion and/or microfluidic cultures[19,31,32]. Microfluidic technologies can considerably enhance cell culture conditions[33]. First, microfluidics provides exquisite spatial and temporal control of the cell's microenvironment, and proper design may allow faithful recreation of in vivo-like conditions. Microfluidics also allows dynamic culture, with continuous or pulsatile perfusion, and the creation of time-dependent gradients of specific bioactive components. The volumes of fluids used in a microfluidic platform are in the low nanoliter range, which drastically reduces operating costs when expensive culture media or components are required. Thanks to a high level of integration, multiple biological processes can be implemented in a single device and experiments and processes run in parallel allowing high-throughput operation[32]. Finally, liquid handling can be automated, and complex protocols programmed[33].

We developed a microfluidic device containing two independent, perfusable 370 µm deep compartments separated by a porous membrane. On top of the porous membrane, a confluent oviduct epithelial cell layer was grown (apical side of the BOEC), while the basolateral compartment was used to mimic the circulating hormone changes that occur during the peri-ovulation period. The two compartments were designed as rectangles (2800 µm wide × 3000 µm long) to ensure uniform shear stress across the entire epithelial layer under perfusion (5 µl h$^{-1}$). Importantly, the apical compartment contained pillars to trap oocytes and/or embryos. This design permitted the continuous apical perfusion of the oviduct epithelial cell layer, which is required to maintain its functional differentiation, throughout the period of IVF and IVP (Fig. 1). A point considered essential in the design of the oviduct-on-a-chip was the total thickness of the apical compartment of the device, which was not higher than 2 mm to allow live imaging of the epithelial cells, gametes and embryos inside the chip (Supplementary Movie 1). Devices were successfully manufactured from poly(dimethylsiloxane) (PDMS), a fairly inexpensive, transparent, gas-permeable, water-impermeable, copyright-free, and rapidly prototyped elastomeric material[34]. PDMS has previously been successfully utilized to fabricate in vitro embryo culture systems[10,19,33,35,36].

### BOEC morphology, differentiation, and responses to hormones.
Two different flow rates were tested on BOECs: 30 and 5 µl h$^{-1}$ based on literature about perfusion of lung epithelial cells[37]. The higher flow rate was discarded because cells under this condition lost their normal morphology and started blebbing (Supplementary Fig. 1). BOECs attached to and proliferated over the entire apical compartment of the microfluidic device, forming a tight cell monolayer (Supplementary Fig. 2 and Supplementary Movie 2). Moreover, some areas exhibited villus-like structures that resembled mucosal folding of the oviduct in vivo (Supplementary Fig. 2). After addition to the apical culture chamber, sperm cells were found to attach to both ciliated and non-ciliated epithelial cells (Supplementary Fig. 2). A total of three different pools of epithelial cells, and 18 microfluidic devices per pool, were used to investigate: (1) cell confluence via both trans-epithelial electrical resistance (TEER) measurements and an apparent permeability assay (Papp); (2) cell morphology, ciliation and oviductal glycoprotein 1 (OVGP1) expression by immunofluorescence; (3) changes in the transcriptome by RNA-sequencing (Cel-seq II). All measurements were compared for BOECs cultured under three different conditions; no hormonal stimulation, luteal-phase simulation and pre-ovulatory phase simulation via the basolateral compartment of the platform ($n = 6$ devices per condition and pool). Fig. 2a summarizes the times and hormone treatments for each group; the hormone treatments were based on the progesterone and estrogen concentrations measured in the oviduct of cows at different stages of the estrous cycle[38].

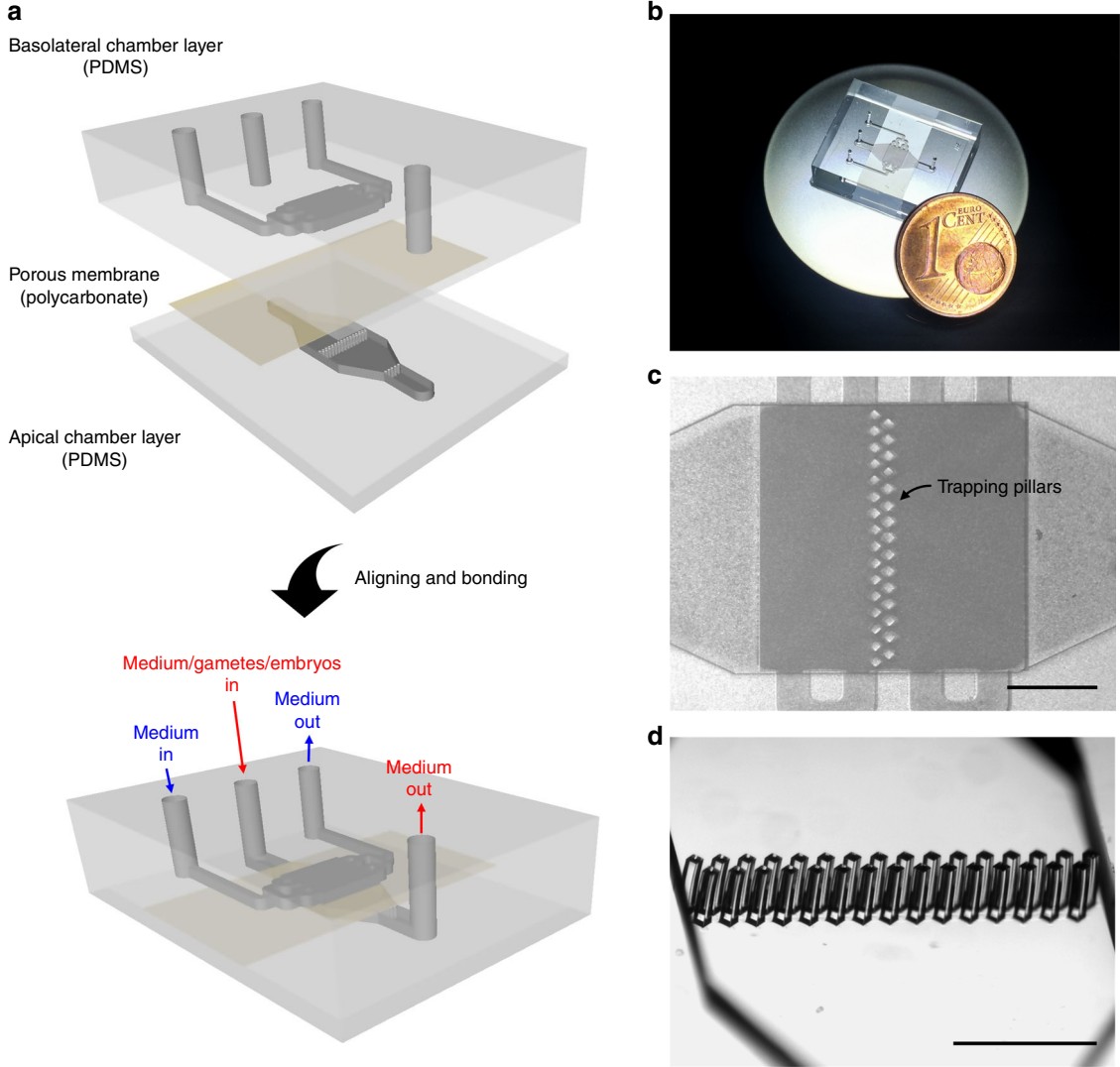

**Fig. 1** Oviduct-on-a-chip platform—design and fabrication. **a** Schematic drawing of the apical and basolateral chambers that are assembled with a porous polycarbonate membrane between them. **b** Picture of an assembled PDMS device. **c** Microscopic picture of the assembled PDMS device, focusing on the apical culture chamber that contains the trapping pillars (TP). **d** Stereomicroscopic picture of the trapping pillars (W 103 µm × L 103 µm × H 380 µm, spacing = 40 µm). Scale bars = 1 mm

TEER measurement is a non-invasive way to assess the confluence and integrity of oviduct epithelial monolayers cultured on a porous substrate[39]. Indeed, TEER measurements are influenced by the expression of specific tight junction proteins, reflecting physical properties of the epithelium[40]. For the oviduct-on-a-chip, the average TEER values of three replicates were (all values are given as mean ± standard deviation): $150.44 \pm 7.14$ ($n = 16$), $186.00 \pm 22.20$ ($n = 18$) and $204.61 \pm 84.50$ ($n = 18$) $\Omega^{-1}$ $cm^{-2}$ for no hormone, the luteal and pre-ovulatory phases, respectively (Fig. 2). The TEER value for the luteal phase was higher than for no hormone ($p < 0.0001$; ANOVA followed by a post-hoc Tukey test), but no statistical difference was observed between no hormone and pre-ovulatory simulation, or between the luteal and pre-ovulatory groups ($p = 0.17$ and $p = 0.81$, respectively; ANOVA followed by a post-hoc Tukey test). The TEER measurements confirmed the formation of a robust epithelial barrier, that also restricted the passage of both fluorescent dextran nanoparticles (4.4 kDa) and fluorescein dye (0.4 kDa) between the basolateral and apical compartments (Fig. 2), mimicking the barrier function of the oviduct epithelium in vivo. Hormone stimulation did not influence the permeability

to the fluorescent dyes (0.4k Da: $p = 0.616$; $p = 0.681$ and $p = 0.994$. 4.4k Da: $p = 0.894$; $p = 0.536$ and $p = 0.809$; for no hormone vs. luteal phase, no hormone vs. pre-ovulatory phase and luteal vs. pre-ovulatory phases, respectively; ANOVA followed by a post-hoc Tukey test). The tight, confluent BOEC monolayers formed in the perfused oviduct-on-a-chip exhibited similar morphology to in vivo oviduct epithelium and, under estrogenic stimulation, produced the major oviductal glycoprotein OVGP1 (Fig. 3a). BOECs cultured inside the chip for 2 weeks under static conditions of both apical and basolateral compartments lost their differentiation and became flat; having an average cell height of $3.8 \pm 0.89$ µm and no cilia. Additionally, after stopping apical compartment perfusion for longer than 3 days, the cells underwent the same loss of differentiation described above. As previously described for a porcine oviductal epithelium[28], stimulation with estrogens to mimic the pre-ovulatory phase increased the height of cultured BOECs ($p < 0.0001$ for all groups comparisons; ANOVA followed by a post-hoc Tukey test; Fig. 3b). Furthermore, hormone stimulation enhanced the number of ciliated cells compared to no added hormones, with no significant difference between luteal and pre-ovulatory phase

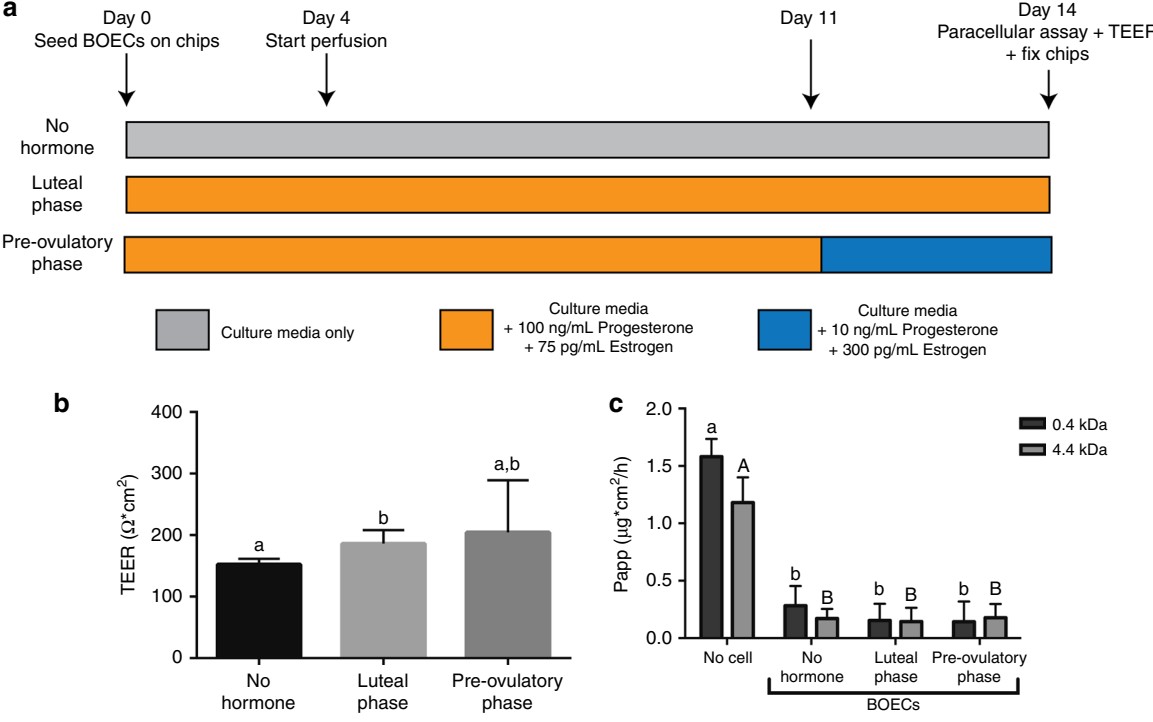

**Fig. 2** Hormonal stimulation experimental design and effects on the trans-epithelial electrical resistance (TEER) and paracellular permeability (Papp). **a** Experimental design for mimicking the luteal and pre-ovulatory phases. **b** TEER measurements in the microfluidic devices under the three conditions; values were adjusted for the resistance found in an empty device ($F = 14.503$, $p < 000.1$; ANOVA followed by a post-hoc Tukey test). **c** Apparent permeability (Papp) of 0.4 and 4.4 kDa fluorescent markers in devices without cells (no cell) or in the presence of bovine oviductal epithelial cells (BOECs) under the three experimental conditions ($F = 0.537$, $p = 0.590$ and $F = 0.583$, $p = 0.564$ for 0.4 and 4.4 kDa, respectively; ANOVA followed by a post-hoc Tukey test). Graphs of **b** and **c** display average + s.d. Different letters (a vs b in **b**; a vs b and A vs B in **c**) indicate statistically significant differences ($n = 6$ devices per condition and pool, $p < 0.05$)

simulation ($p = 0.014$; $p = 0.002$ and $p = 0.172$; for no hormone vs. luteal phase, no hormone vs. pre-ovulatory phase and luteal vs. pre-ovulatory phase, respectively; ANOVA followed by a post-hoc Tukey test; Fig. 3c). Additionally, as described previously for porcine, human and canine oviduct[28,32,41], the pre-ovulatory phase (high estrogen) enhanced OVGP1 expression compared to control or luteal-phase conditions ($p = 0.829$; $p = 0.002$ and $p < 0.0001$; for no hormone vs. luteal phase, no hormone vs. pre-ovulatory phase and luteal vs. pre-ovulatory phase, respectively, ANOVA followed by a post-hoc Tukey test; Fig. 3d). Note that the PDMS material has hydrophobic properties[42] and has been reported to absorb steroids[43]. However, after imposing changes in steroid levels in the perfusion medium (at perfusion rates of 5 μl h$^{-1}$) within 24 h the collected perfusion medium that passed the outlet of 5 cell-free PDMS chips showed a nearly identical steroid levels as what was perfused through the inlet (Supplementary Fig. 3). Note that some PDMS absorption of the 100 ng ml$^{-1}$ progesterone used in the perfusion medium was observed in the first 12 h while this absorbed progesterone was not released during subsequent perfusion with progesterone free medium. However, in general the hormonal switches imposed by the perfusion medium were effective within 12–24 h in these PDMS chips which make them suitable to mimic physiological occurring changes in steroid levels at the peri-ovulatory timescale.

To evaluate the effects of steroid hormone treatment (luteal and pre-ovulatory phase simulation) on transcriptional activity in the epithelial cells, we performed RNA-sequencing (RNA-seq). A total of 14,383 genes were detected by Cel-seq II, with no significant difference (fold change 1/False discovery rate < 1%) between no hormone stimulation (CNH) and the simulated luteal phase (CP). By contrast, 183 transcripts were upregulated and 140

were downregulated in the pre-ovulatory phase (CE) compared to the CP. Functional gene ontology (GO) clustering of upregulated genes into "molecular and biological processes" indicated an increase in genes related to ciliogenesis and cilia movement in the pre-ovulatory phase (Fig. 4a), as well as an increase on estrogen related receptor alpha (ESRRA). Progesterone has previously been reported to inhibit oviduct epithelial cell cilia beating in man, mouse, guinea pig and cow[44–47]. The pre-ovulatory phase also showed increased expression of transcripts related to the immune response (Fig. 4a) similar to what has previously been described in vivo[48]. The oviductal epithelium must presumably protect itself from any pathogens that may accompany spermatozoa and seminal fluids. The ovarian steroid hormone-dependent change in immune responsiveness is likely a physiologically important process activated during the pre-ovulatory phase, when spermatozoal contact is expected. Other upregulated GO pathways in the pre-ovulatory phase include; inflammatory response, regulation of protein activation cascade, regulation of protein processing and maturation, retinoid metabolic process, and regulation of endocytosis (Fig. 4a). The luteal-phase epithelium was characterized by increased cell-cell junction organization, response to growth factors, antioxidant activity, lipid biosynthetic and metabolic processes, response to oxidative stress, epithelial cell proliferation and regulation of chemotaxis as well as an increased expression of progesterone receptor membrane components 1 and 2 (PGRMC1 and PGRMC2) (Fig. 4b; see Supplementary Data 1 for a complete list of GO pathways differentially regulated between the pre-ovulatory and luteal phases)

BOECs cultured in the oviduct-on-a-chip, independent of hormone stimulation, expressed genes related to sperm-oviduct adhesion[49] (FUCA1, ANXA1, ANXA2, ANXA4 and ANXA5),

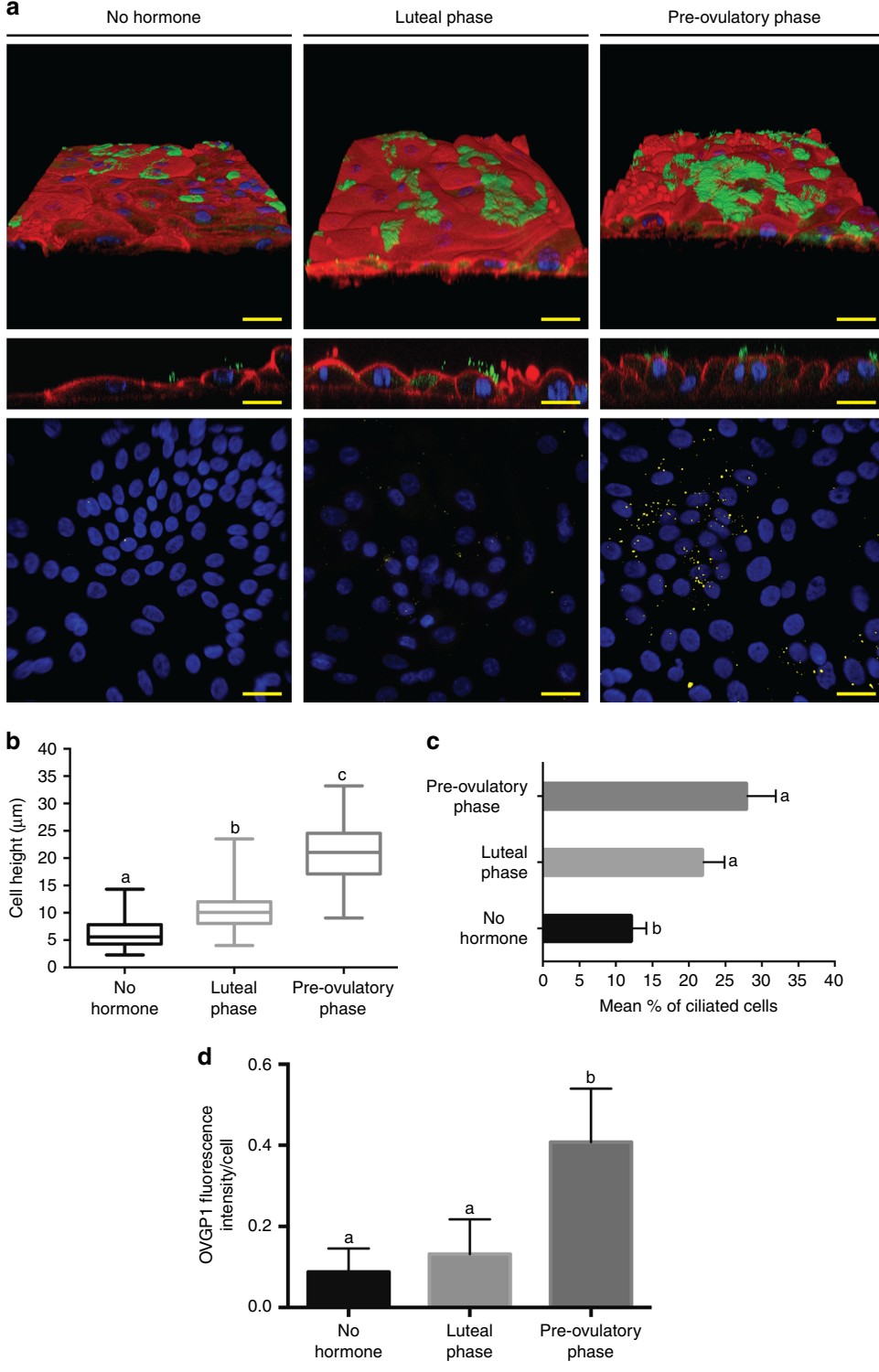

**Fig. 3** Effects of hormone stimulation of 3D-cultured BOECs on cell height, ciliation and oviductal glycoprotein 1 (OVGP1) expression under control, and simulated luteal and pre-ovulatory conditions. **a** Top and middle figures: 3D reconstruction of confocal immunofluorescent (IF) images for cilia (acetylated alpha-tubulin, green), nuclei (HOECHST 33342, blue), and actin filaments (phalloidin, red); bottom figures: IF for nuclei (blue) and OVGP1 (yellow). **b** Quantification of cell height in the different groups ($F = 697.51$, $p < 0.0001$; ANOVA followed by a post-hoc Tukey test). **c** Average percentage of ciliated cells for each group ($F = 20.415$, $p = 0.002$; ANOVA followed by a post-hoc Tukey test). **d** Quantification of OVGP1 expression adjusted for cell number ($F = 12.52$, $p < 0.0001$; ANOVA followed by a post-hoc Tukey test). Graph of **b** displays average ± s.d and graphs of **c** and **d** display average + s.d. Different letters (a vs b vs c in **b**; a vs b in **c** and **d**) indicate statistically significant differences ($n = 6$ devices per condition and pool, $p < 0.05$). Scale bars represent 10 μm

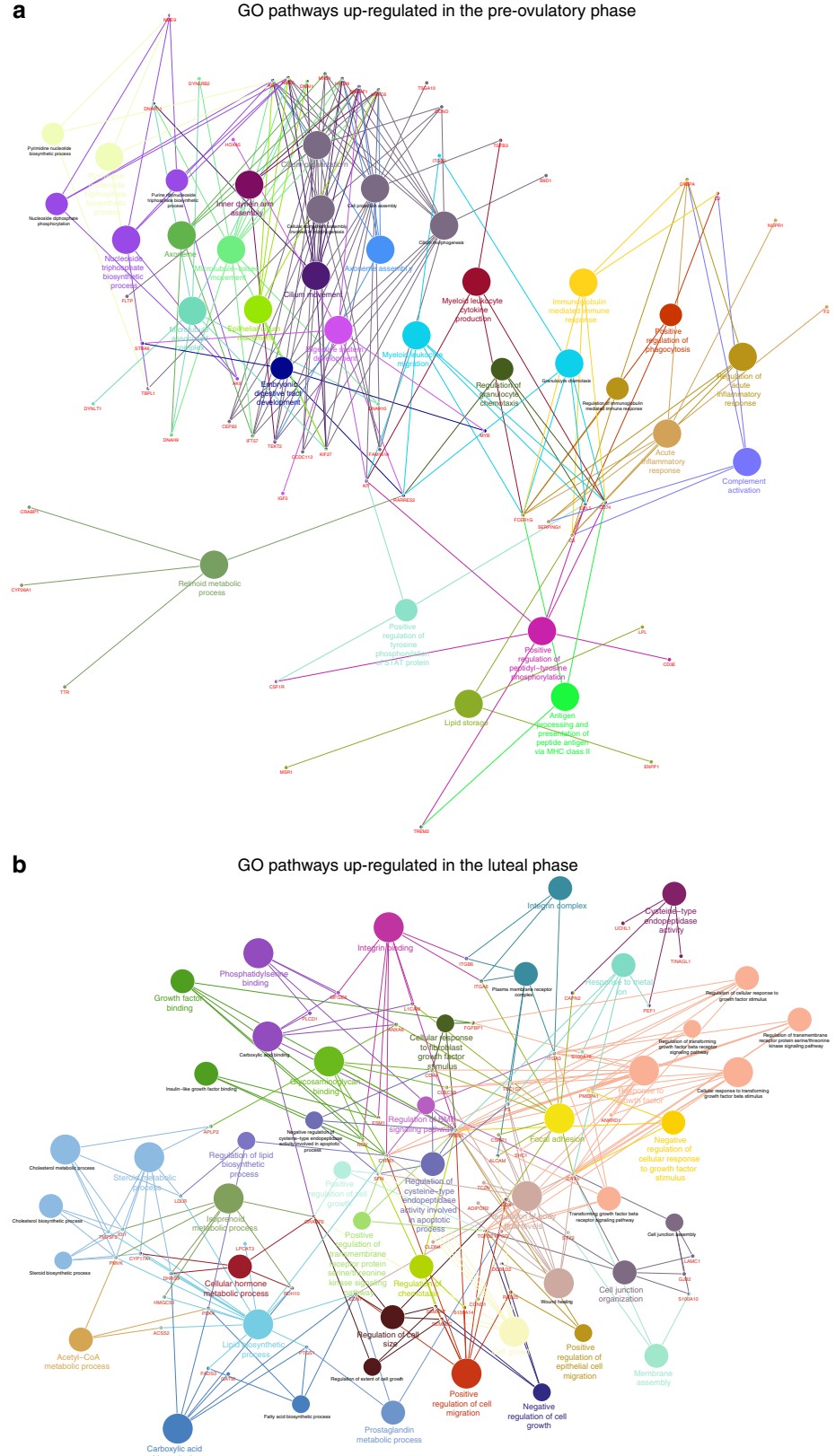

**Fig. 4** Functionally grouped gene ontology (GO) terms for upregulated or downregulated gene expression in simulated pre-ovulatory and luteal phases. The CytoScape plugin ClueGO was used to group the genes into functional GO terms of "molecular processes" and "biological processes". **a** Upregulated GO pathways in the pre-ovulatory phase; and **b** upregulated GO pathways in the luteal phase

COC-oviduct interaction[50] (*MUC20, SPP1, PDGC* and *CSTA*), fertilization[50,51] (*HEXDC, HEXIM1, HYAL2, GLB1, HSPA9, HSPA8, HSP90AB1,RPS6, CD46, CD9, MFGE8, ADAM9,* and *NTS*) and embryo development[50,52] (*C3, IGF2, TGFB2,* and *TGFB3*). Together, the expression of these genes in the oviduct-on-a-chip supports the conclusion that the platform permits the in vitro culture of a functional bovine oviduct epithelium that responds appropriately to ovarian steroid hormones.

**The oviduct-on-a-chip supports IVF and embryo development**. Using in vitro matured (IVM) oocytes, in vitro fertilization (IVF) was performed either in a 4-well dish (in vitro embryos—VT) or inside a microfluidic device containing a confluent layer of differentiated BOECs (on chip embryos—CH, Supplementary Movie 3). Ten devices (from the three different animal pools) were used for on chip fertilization and culture. In the oviduct-on-a-chip device, both first cleavage and 8–16 cells formation were observed. However, neither cleavage (56.0% vs. 84.4%, $p = 0.0021$; ANOVA followed by a post-hoc Tukey test) nor 8–16 cells formation (36.7% vs. 53.7%, $p = 0.0089$; ANOVA followed by a post-hoc Tukey test) was as successful on chip as in an optimized in vitro embryo production protocol. This reduced success can in part be explained by the fact that nearly half of the mature oocytes/embryos (103 out of 230) were able to 'escape' through the pillars and were subsequently either lost during perfusion or became trapped between the pillars, which resulted in developmental arrest (Supplementary Fig. 4). Another factor that influences embryo development is shear stress. Previous studies have shown that high shear stresses can impair mouse embryo development[53] through the activation of stress-activated protein kinase-mediated apoptosis, and that early stage embryos (8–16 cells) are more sensitive to shear stress than blastocysts[53]. In our experiments, the average shear stress exerted on the embryos was $0.70 \pm 0.46$ dyne cm$^{-2}$. However, embryos trapped between pillars and other lines of embryos, were exposed to a maximum shear stress of 2.06 dyne cm$^{-2}$ (Fig. 5), which is higher than the values shown to have a negative impact on mouse embryos (1.2 dyne cm$^{-2}$).

**Global methylation of on chip are similar to in vivo zygotes**. The global methylation patterns of 30 in vitro (*VT*), 30 on chip (*CH*) and 30 in vivo (*VV*) zygotes were analyzed using fluorescent 5mC staining (Fig. 6), with the fluorescence intensity being normalized to that of a general DNA stain (propidium iodide: PI). Zygotes were analyzed independent on their developmental stage. We found that nuclear intensity of 5mC of *VT* was 4.7-times higher than in *VV* ($p = 0.014$; ANOVA followed by a post-hoc Tukey test) and 2.6-times higher than in *CH* zygotes ($p = 0.028$). Interestingly, the global methylation staining intensity did not differ between *VV* and *CH* zygotes ($p = 0.876$; ANOVA followed by a post-hoc Tukey test). These results collectively suggest that the interaction between the gametes and/or zygotes with the epithelium in the oviduct-on-a-chip platform overcomes the changes to the demethylation process that results during standard in vitro culture. Similar failure of pronucleus demethylation during ARTs has been reported for porcine zygotes, where the effect was most marked after conventional IVF and slightly less pronounced after parthenogenetic activation or somatic cell nuclear transfer[16]. Likewise, partial recovery of the methylation levels at the blastocyst stage was observed in pig embryos cultured in the presence of female reproductive tract fluids (oviductal and uterine fluids)[54].

**Zygote transcriptome changes in different systems**. Here, we used Cel-seq II to compare the transcriptome of individual bovine zygotes produced under different conditions: in vivo (*VV*), in vitro (*VT*), and on chip (*CH*) ($n = 10$ zygotes for each group). A total of 18,258 transcripts were detected, of which 14,042 were common to *VV*, *VT* and *CH* zygotes. A principal component analysis (PCA) revealed two distinct clusters of zygotes: Group 1 (G1) contained all *VT*, two *VV* and five *CH* zygotes; and Group 2 (G2) comprised eight *VV* and five *CH* zygotes (Fig. 7). In G1, 3,063 transcripts were upregulated and 3,507 downregulated compared to G2 (see Supplementary Data 2 for all differentially expressed genes). From the downregulated transcripts, four important GO pathways were identified: initiation of transcription, initiation of translation, (de)methylation and (de)acetylation (Table 1, Supplementary Figs. 5–7). This indicates that zygotes in G1 have a delayed minor embryonic transcriptome activation compared to zygotes in G2. Likewise, the oviduct epithelium has an important role in regulating embryo development, since all zygotes that were not in contact with oviduct (*VT* zygotes) were in the delayed group whereas 80% of *VV* zygotes were in G2.

The oviduct-on-a-chip platform rescued the gene expression pattern of half of the analyzed zygotes. By contrast, the other half of the *CH* zygotes clustered with the G1 delayed zygote group, which also included 20% of the *VV* zygotes and all *VT* zygotes. One possible explanation for the presence of *CH* and *VV* zygotes in the delayed G1 group is that oocyte penetration and/or activation was not simultaneous. We used transvaginal endoscope-guided oviduct flushing to collect *VV* zygotes 43–47.5 h post insemination (hpi; 19–23.5 h post presumed ovulation), while *VT* and *CH* zygotes were collected 20–22 h after incubation with sperm cells. Although embryos were collected at similar times after sperm–oocyte encounter, we were not able to distinguish different pronuclear stages of the zygotes collected (bovine zygotes have dark cytoplasm, which prevents assessment of the pronuclei by normal light microscopy as performed in mouse and human zygotes). Therefore, zygotes were selected purely on the basis of two extruded polar bodies, which may have allowed for asynchrony to affect zygote stage.

## Discussion

"ART in humans is a multibillion-dollar industry, full of eager patients and a contradictory scientific literature full of vague concerns"[55]. As a consequence, the majority of ART research has focused on improving the chances of producing a baby, but has neglected the potential long-term impact of ART on the health of the newborns[55]. In mice and other animal models, the possible effects of ART on offspring development and health have been investigated (for review see Feuer & Rinaudo[56]). However, mouse data is of limited utility to human embryogenesis because of large differences in gene expression patterns and genome sequences. Indeed for these aspects, human embryos are more similar to bovine embryos[57]. Bovine and human preimplantation embryos have also been reported to be similar in terms of biochemical and intrinsic paternal and maternal regulatory (imprinting) processes[58]. Along with the ethical issues of experimenting on human embryos, all these reasons justify the use of bovine oocytes/embryos as a model for human embryogenesis.

In a previous study, we demonstrated benefits of the oviductal environment to support fertilization[19]. However, our first oviduct-on-a-chip platform did not allow perfusion during embryo culture. Additionally, the material used to produce the original devices released toxic compounds, which adversely affected the developing embryos while PDMS did not[20]. Therefore, we developed a PDMS based platform that promoted cell growth and differentiation under perfusion, and that allowed live imaging and embryo production. BOECs grown in the oviduct-on-a-chip responded to steroid hormone simulation of the luteal

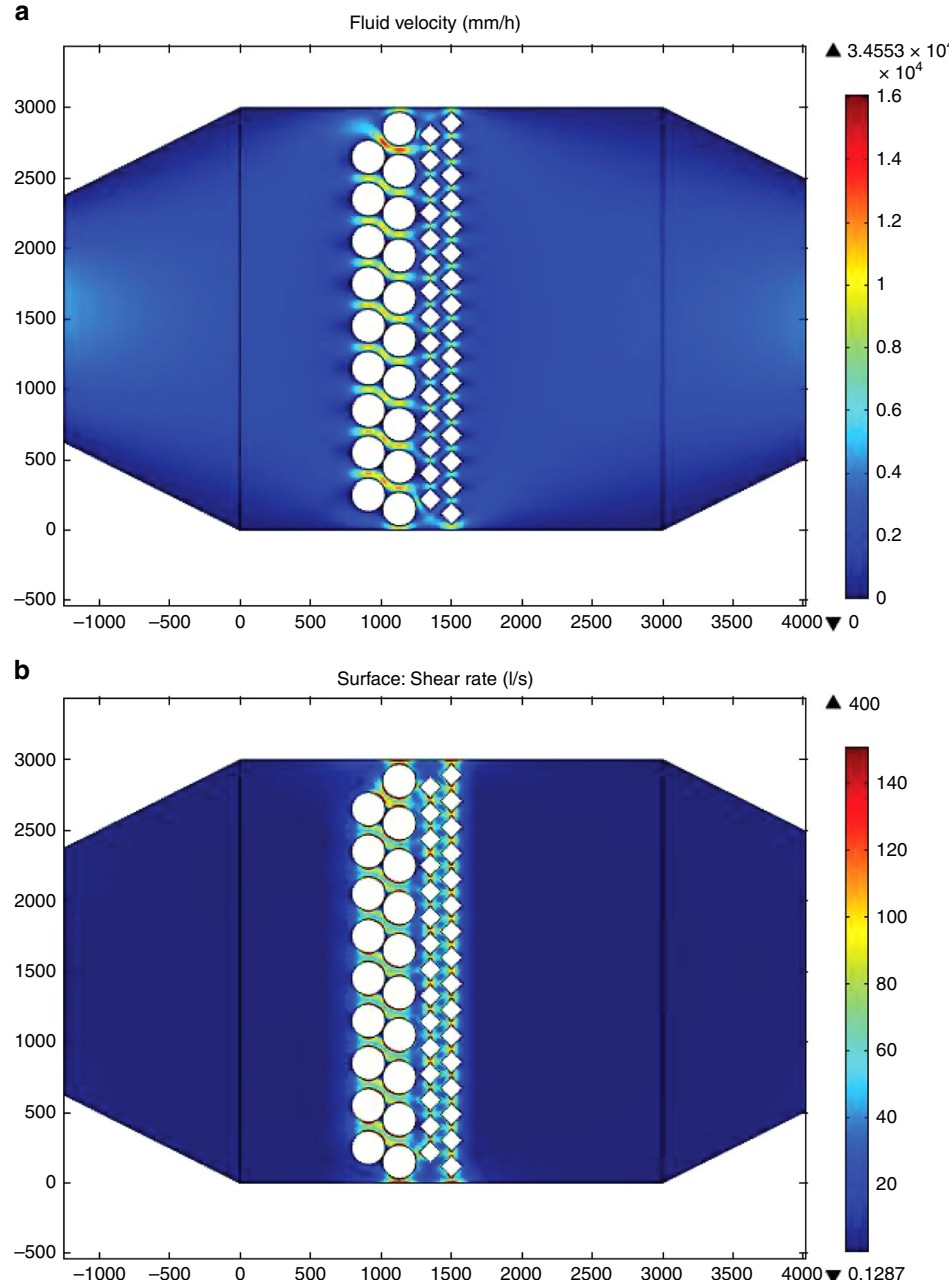

**Fig. 5** Modulation of the flow and shear rate inside the oviduct-on-a-chip. In **a** notice the evenly distributed flow, that is direct and increases between pillars and "COCs/embryos" (white circles), mimicking IVF simultaneous with perfusion of the apical compartment. In **b** simulation of shear rate, note that embryos between the pillars and other embryos experience a higher shear rate

and pre-ovulatory phases. Transcriptome changes similar to the in vivo luteal phase were observed after progesterone treatment, included reduced expression of genes involved in ciliary activity, and increases in those involved in tight junction formation and transmembrane signaling receptor activity. By contrast, a high estrogen environment increased expression of genes related to the immune response, regulation of protein processing, maturation and cell projection morphogenesis[48]. These results collectively demonstrate that the oviduct-on-a-chip allowed BOEC growth and differentiation similarly to that observed in vivo. Furthermore, the BOEC monolayer exhibited villus-like structures that resembled natural oviduct folding[25]. The oviduct-on-a-chip also supported fertilization and embryo development up to the 8–16 cells stage, although 8–16 cells production rates were not as high as for optimized IVP protocols. We conclude that the chip could

be further improved by: (1) minor changes to its design to ensure that COCs/embryos are retained during perfusion; (2) mimicking the steroid hormone environment of the peri-conception period; and (3) analyzing and optimizing flow rates and shear stress to better protect developing on chip embryos.

Although reduced cleavage and 8–16 cells formation rates were observed, on chip (*CH*) zygotes were more similar to in vivo (*VV*) than to conventional in vitro (*VT*) zygotes in terms of their global DNA methylation levels and transcriptome. Interestingly, *VV* and *CH* zygotes exhibited lower global DNA methylation than *VT* zygotes, which is presumably related to the higher expression of genes involved in (de)methylation (*DNMT3b, DNMT1, TET1, TDG, TRIM28, KDM6A, APEX1* and *DDX5*) in 80% of the *VV* and 50% of the *CH* zygotes (G2). This lower methylation level seems to be essential for the minor (zygotic) genome activation,

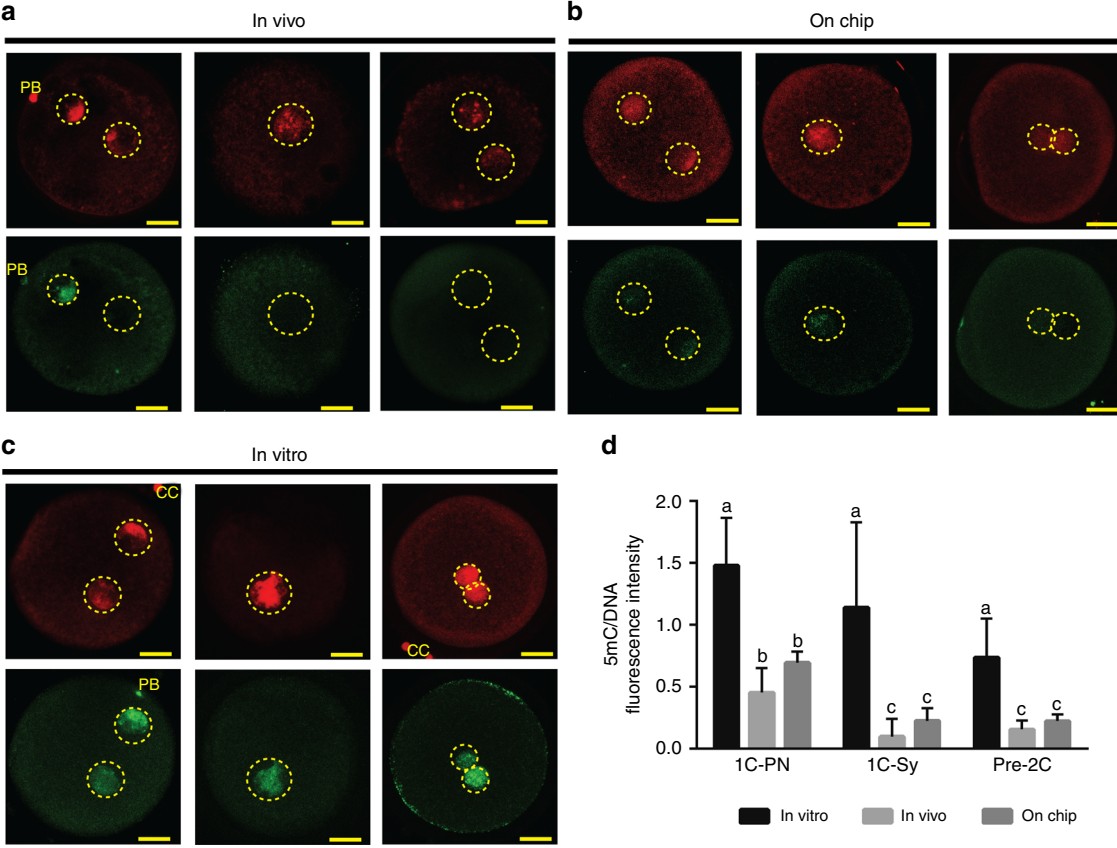

**Fig. 6** In vivo (**a**), on chip (**b**), and in vitro (**c**) zygotes; indirect immunofluorescent staining for global methylation (5mC, green) and DNA (propidium iodide, red) in the pronuclei. **d** Quantification of 5mC staining in zygotes, 5mC fluorescence intensity was normalized using total DNA fluorescence ($n = 30$ for in vivo, $n = 30$ for in vitro and $n = 30$ for on chip zygotes; $F = 7.458$, $p = 0.011$; ANOVA followed by a post-hoc Tukey test). Graph of **d** display average + s.d. Negative controls omitting primary antibody were used and no fluorescence was observed. Different letters (a vs b vs c in **d**) indicate statistical differences between the groups ($p < 0.05$). Scale bars = 25 um

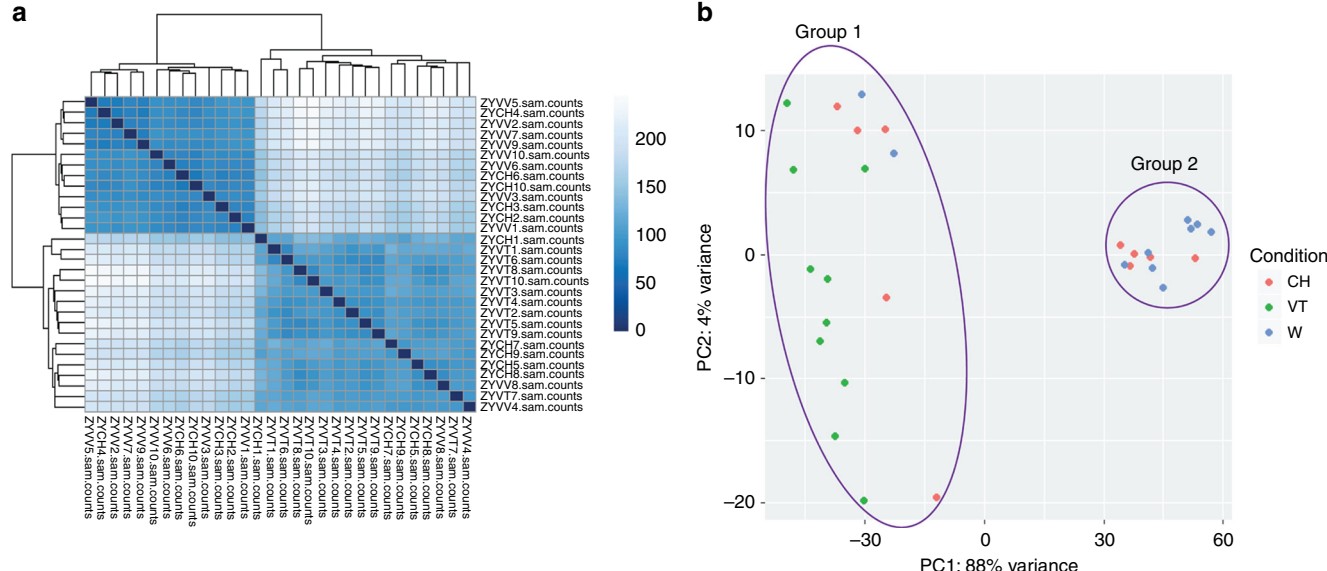

**Fig. 7** Comparison of the transcriptomes identified by Cel-seq for in vivo (*VV*), in vitro (*VT*) and on chip (*CH*) zygotes. **a** Heat map comparing all zygotes. **b** Principal component analysis (PCA) of the transcriptomes for in vivo (*VV*, blue), in vitro (*VT*, green) and on chip (*CH*, red) zygotes; PC1 and PC2 represent the top two dimensions of the differentially expressed genes among the zygote groups. Note, from **a** and **b**, the division between two main clustering groups, Groups 1 and 2 in (**b**)

**Table 1 Upregulated and downregulated genes related to DNA (de)methylation and histone (de)acetylation processes between G2 and G1 zygotes**

| Gene | Log2 fold change | p-value |
|---|---|---|
| TET1 | 7.17 | 4.73E−79 |
| IDH1 | 4.33 | 8.14E−123 |
| HDAC6 | 3.92 | 1.03E−20 |
| TDG | 3.55 | 2.31E−19 |
| DNMT3b | 3.45 | 2.23E−25 |
| HDAC5 | 3.41 | 2.28E−19 |
| HDAC8 | 3.27 | 1.56E−27 |
| IDH2 | 3.25 | 2.88E−13 |
| AMPD2 | −1.58 | 0.0135 |
| HDAC7 | −1.58 | 2.67E−05 |
| MBD4 | −2.38 | 1.55E−18 |
| DNMT1 | −2.67 | 1.37E−25 |
| HDAC9 | −3.49 | 0.0007 |
| HDAC11 | −3.93 | 1.14E−14 |
| TET3 | −4.72 | 4.62E−13 |

since an upregulation of genes related to transcription and translation initiation was apparent in G2 compared to G1 zygotes. In the current study, we did not measure differences following the major wave of epigenetic reprogramming, which is more complex and takes place at a later stage of pre-implantation embryo development primarily within the uterine environment. In addition, the changes in global DNA methylation status were detected using an immunofluorescence labeling method, which provides an indirect index of DNA methylation status and depends on the specificity and affinity of the antibodies used; bisulfite sequencing for key imprinted genes or more direct techniques for investigating the DNA methylome in zygotes and later stage embryos may reveal how the observed differences in (epi)genome at the zygote stage relate to embryo developmental competence and epigenetic reprogramming following embryonic genome activation but before implantation. Despite of the absence of post-genome activation data, when we compared our data with genes described to be first expressed at the 4, 8, 16-cell or blastocyst stages of *IVP* bovine embryo development[59], 24% of the 220 genes reported to be detected at only one of these stages, were upregulated in the G2 zygotes. This suggests that standard IVP conditions delay zygote transcriptome activation, but that the delay can be ameliorated using our oviduct-on-a-chip platform. Overall, our results highlight the importance of a more in vivo-like environment when studying pathways related to normal fertilization and zygote formation in vitro. Future studies should focus on the relevance of this improved environment for further (epi)genetic reprogramming events in developing embryos, when the use of in vivo embryos is not an option for ethical reasons. The addition of oviductal and uterine fluids to culture medium was recently described to 'improve' blastocyst gene expression and DNA methylation patterns in porcine embryos[54]. However, the reported recovery of in vivo characteristics was only partial whereas our oviduct-on-a-chip yielded 50% of zygotes with no discernible difference in gene expression pattern to *VV* zygotes. It is therefore possible that not only oviduct epithelial secretions, but also direct contact with the epithelial cells influences the embryonic transcriptome and epigenome. In support of this theory, the apposition of blastocysts to endometrial cells, but not contact with endometrial secretions, was able to initiate trophectoderm differentiation in mouse embryos[60].

In conclusion, we have designed a tool for investigating early maternal-gamete/embryo interaction in which we can produce zygotes that closely resemble in vivo zygotes. Using this state-of-the-art oviduct-on-a-chip platform, we expect to increase our overall understanding of gamete interaction, fertilization and early embryo development, by more faithfully mimicking the in vivo environment. In contrast to previously described micro-fluidic models[32], we used cell rather than tissue culture, which has several advantages. First, the apical and basolateral compartments were completely separated, which allows distinct collection of secreted factors from, or introduction of exogenous factors to, the apical (luminal) and basolateral (blood circulation) compartments. Both culture conditioning and the introduction of estrous cycle changes allow creation of an even more in vivo-like environment, which is of interest when testing or developing new IVP supplements or when performing toxicological assays. Second, gene-editing of the BOECs (such as by CRISPR/Cas9) is conceivable using this approach, for instance to investigate the effects of specific oviductal factors on gametes or embryos, or by using the model to edit embryo genomes, for instance with Genome-editing via Oviductal Nucleic Acids Delivery (GONAD)[61]. This would help reduce the need for animal experimentation and, in particular, mouse knockout models.

Beyond its use for refining ART, the oviduct-on-a-chip platform could find other exciting applications. Since it permits live imaging for tracking cell migration and/or specific molecular pathways, it opens avenues for interrogating pathways associated with tubal derived ovarian cancers and thereby for the identification of biomarkers for the early diagnosis of this lethal disease. Ultimately, the oviduct-on-a-chip platform could facilitate development of patient-derived in vitro cancer models, which could be extremely valuable for personalized medicine purposes.

## Methods

**Chemicals**. Unless stated otherwise, all chemicals were obtained from Sigma Chemical Co. (St. Louis, MO) and were of the highest available purity.

**Design and fabrication of the oviduct-on-a-chip**. The microfluidic devices (Fig. 1) were fabricated using soft lithography[62]. Uncured PDMS mixture (GE RTV-615, Permacol B.V., Ede, The Netherlands; prepolymer:curing agent = 7:1) was poured on 4′-silicon wafers with 380 µm thick patterns of SU-8 100 (Micro-Chemicals GmbH, Ulm, Germany) and cured for 60 min at 80 °C. The apical and basolateral compartments were peeled off the mold, and holes for inlets and outlets were made using a 25-gauge punch (Syneo Co., Angleton, TX, USA). A 10-µm thick porous polycarbonate membrane (TRAKETCH® PC10, pore size: 0.4 µm, pore density: $100 \times 10^6$ cm$^{-2}$, SABEU GmbH & Co. KG) was sandwiched between the aligned apical and basolateral layers and bonded using PDMS mortar[63]. Before use with cells, the chambers were sterilized for 1 h in 70% ethanol, washed three times for 30 min each in phosphate-buffered saline (PBS; 163.9 mM Na$^+$, 140.3 mM Cl$^-$, 8.7 mM HPO$_4$$^{3-}$, 1.8 mM H$_2$PO$_4$$^-$, pH 7.4; Braun, Melsungen, Germany) and washed overnight in HEPES-buffered Medium 199 (Gibco BRL, Paisley, U.K.) supplemented with 100 U ml$^{-1}$ penicillin and 100 µg ml$^{-1}$ streptomycin (Gibco BRL, Paisley, U.K.). The porous membrane was finally coated with a Matrigel solution (3 µg ml$^{-1}$ in DMEM/F12; Corning, USA) at 37 °C for 2 h. These pre-treated cell-free chips were also used to detect steroid absorption and releasing properties of the PDMS material. See Supplementary Methods for chip-tubing assembling and pump connection.

**Computational fluid dynamics and shear stress simulation**. Computation of the flow and shear stresses in the apical compartment was performed using the CFD mode of the commercial finite element code COMSOL Multiphysics 4.4 (COMSOL Inc., MA, USA). To simulate velocity within the microfluidic channel, the "Steady Flow" module was used with liquid set to water and a flow rate of 5 µl h$^{-1}$. The shear stress ($\tau$) within the fluid channel is related to the volume flow rate ($Q$), the fluid viscosity ($\eta$), and the channel dimensions (height $h$ and width $w$) as follows:

$$\tau = 12\left(\frac{Q\eta}{h^2 w}\right) \qquad (1)$$

**Isolation of oviduct cells and cell culture**. Cow oviducts were collected from a local abattoir immediately after slaughter and transported to the laboratory on ice, within two hours. The oviducts were dissected free of surrounding tissue and washed three times in cold PBS supplemented with 100 U ml$^{-1}$ penicillin and 100 µg ml$^{-1}$ streptomycin. BOECs were isolated by squeezing the oviductal contents out of the ampullary end of the oviducts, and collected in HEPES-buffered Medium 199 supplemented with 100 00 U ml$^{-1}$ penicillin and 100 µg ml$^{-1}$ streptomycin.

The cells were washed twice by centrifuging for 500×*g* for 5 min at 25 °C in HEPES-buffered Medium 199 supplemented with 100 U ml$^{-1}$ penicillin and 100 µg ml$^{-1}$ streptomycin. The cells were then cultured for 24 h in HEPES-buffered Medium 199 supplemented with 100 U ml$^{-1}$ penicillin, 100 µg ml$^{-1}$ streptomycin, and 10% fetal calf serum (FCS; Bovogen Biologicals, Melbourne, Australia). During these 24 h, the cells formed floating vesicles with outward beating cilia. These vesicles were collected, centrifuged at 500×*g* for 5 min at 25 °C, suspended in DMEM/Ham's F12 medium (DMEM/F12 Glutamax I, Gibco BRL, Paisley, U.K.) supplemented with 5 µg ml$^{-1}$ insulin, 5 µg ml$^{-1}$ transferrin, 10 ng ml$^{-1}$ epidermal growth factor, 50 nM trans-retinoic acid, 10 mM glutathione, 100 µg ml$^{-1}$ gentamycin, 5% FCS and 2.5 mg ml$^{-1}$ amphotericin B (chip culture medium, adapted from Ferraz et al.[19]), and pipetted up and down several times to mechanically separate the cells. Next, cells from three different donor animals were mixed and seeded into the apical compartments of the oviduct-on-a-chip devices at a concentration of 10 × 10$^6$ cells ml$^{-1}$ (17.8 × 10$^6$ cells cm$^{-2}$) and allowed to attach and reach confluence during 4 days under static conditions. The culture medium in the basolateral compartment was manually replaced twice a day during the first 4 days during which the device was kept in a humidified atmosphere of 5% CO$_2$ and 7% O$_2$ at 38.5 °C. Once the cells had reached confluence (4 days after seeding), both the basolateral and apical compartments were maintained under constant flow perfusion (5 µl h$^{-1}$) using a Programmable Aladdin Syringe Pump (WPI, Germany), in a humidified atmosphere of 5% CO$_2$, 7% O$_2$, and 38.5 °C.

**Hormonal stimulation.** Cultures were stimulated periodically with exogenous progesterone (P4) and estradiol 17β (E2) via the basolateral medium. The concentrations of E2 and P4 were based on in vivo oviductal fluid concentrations reported for cows[38]. From the day they were seeded into the chips (day 0), BOECs were cultured under one of three different conditions: (1) a control with no hormone stimulation in which the basolateral channel was perfused with chip culture medium plus 1% ethanol for 14 days; (2) a simulated luteal phase in which the basolateral channel was perfused with chip culture medium supplemented with 100 ng ml$^{-1}$ P4 and 75 pg ml$^{-1}$ E2 for 14 days; and (3) a simulated pre-ovulatory phase in which the basolateral channel was perfused with chip culture medium supplemented with 100 ng ml$^{-1}$ P4 and 75 pg ml$^{-1}$ E2 for 11 days followed by 10 ng ml$^{-1}$ P4 and 300 pg ml$^{-1}$ E2 for 3 days (Fig. 2). In a control experiment, the medium with 100 ng ml$^{-1}$ P4 and 75 pg ml$^{-1}$ E2 was perfused for 60 h in cell-free chips (*n* = 5) followed with a perfusion in steroid free medium for 60 h. During the entire perfusion period the apical compartment fluid movement was stopped. At intervals the basolateral flow through fluid was collected and progesterone levels were assessed using solid-phase [$^{125}$I] RIA (Coat-A-Count; TKPG; Siemens Medical Solutions Diagnostics, Los Angeles, CA, USA) according to manufacturer with slight modifications[64] as well as on both perfusion fluids.

**Paracellular tracer flux assay.** For permeability measurements, 8 µl of a dextran–TRITC (4 kDa) or fluorescein disodium salt (0.4 kDa) solution in culture medium (48 µg ml$^{-1}$) was perfused through the apical channel on day 14 of culture, while unsupplemented culture medium was perfused through the basolateral compartment. Two hours later, the fluorescence intensity was measured in the medium recovered from the basolateral chamber of individual devices. An empty device without any BOECs served as a control. The fluorescence intensity was measured using a BMG Clariostar fluorimeter (Ortenberg, Germany). The apparent permeability Papp (µg cm$^2$ h$^{-1}$) was calculated using the following formula:

$$\mathrm{Papp} = (Q/t)/\left(\frac{1}{A}\right) \qquad (2)$$

Where *Q*/*t* is the steady-state flux (µg ml$^{-1}$ h$^{-1}$) and *A* the total area of diffusion (cm$^2$).

**Trans-epithelial electrical resistance (TEER).** TEER measurements were performed on day 14 of culture. Two Ag/AgCl wire electrodes (World Precision Instruments, Germany) were sterilized for 10 min in 70% ethanol and connected to a digital volt-ohm (Millicell, USA) using alligator clips. The microfluidic devices were filled with HEPES-buffered Medium 199 supplemented with 100 U ml$^{-1}$ penicillin and 100 µg ml$^{-1}$ streptomycin injected into the apical and basolateral compartments through silicone tubing connected to the inlet ports. Electrodes were inserted into each compartment (one via the apical and one via the basolateral inlet tubing)[37]. After 1 min of stabilization, the electrical resistance was recorded. The electrical resistance of a blank (device without cells) was measured in parallel. To obtain the TEER measurement (in Ω$^{-1}$ cm$^{-2}$), the blank value was subtracted from the total resistance of the sample, and the final unit area resistance (Ω$^{-1}$ cm$^{-2}$) was calculated by multiplying the sample resistance by the effective area of porous membrane onto which the cells are grown (0.09 cm$^2$).

**Cell ciliation and morphology.** At day 14 of culture, two oviduct-on-a-chip devices were fixed per pool (3 pools, *n* = 6 devices per condition) to assess cilia formation and the morphology of epithelial cells using immunofluorescent staining[19]. Chips were fixed in 4% paraformaldehyde for 30 min, and permeabilized for

30 min using 0.5 % Triton-X100 in PBS. Non-specific binding was blocked by incubation for 1 h in PBS containing 5% normal goat serum at room temperature. The chips were then incubated overnight at 4 °C with rabbit anti-acetylated α-tubulin (1:100, ab125356, Abcam, Cambridge, UK) and mouse anti-OVGP1 (1:50; sc-377267 Santa Cruz Biotechnology, Santa Cruz, CA) primary antibodies. Next, the chips were washed and incubated with an Alexa 488 conjugated goat anti-rabbit antibody and an Alexa 647 conjugated goat anti-mouse antibody (1:100; Santa Cruz Biotechnology, Santa Cruz, CA) for 1 h. Hoechst 33342 (5 µg ml$^{-1}$) was used to stain cell nuclei and phalloidin conjugated to Alexa 568 (1:100) was used to stain actin filaments. For imaging, laser scanning confocal microscopy using a TCS SPE-II system (Leica Microsystems GmbH, Wetzlar, Germany) attached to an inverted semi-automated DMI4000 microscope (Leica) with a ×40 NA 1.25 objective was used. 3D images of the cell monolayers were re-constructed from 0.2 µm Z-stacks using ImageJ software (National Institutes of Health, Bethesda, MD, USA) to evaluate cell morphology, ciliation and OVGP1 expression. A total of six randomly selected areas were imaged per device. For OVGP1 quantification, images were analyzed by evaluating fluorescence intensity for OVGP1 and DNA in each area using ImageJ software. After maximum projection reconstruction of Z-stacks, the fluorescence intensity of each channel was measured and adjusted for cytoplasmic background. The average intensity of fluorescence for OVGP1 was then normalized by dividing the OVGP1 intensity by Hoechst 33342 fluorescence to normalize for DNA content.

**Sperm preparation for IVF and live cell imaging.** Frozen spermatozoa were thawed at 37 °C for 30 s and the spermatozoa washed by centrifugation at 700×*g* for 30 min through a discontinuous Percoll gradient (GE Healthcare, USA) at 27 °C. The supernatant was removed and the pellet suspended in fertilization medium (modified Tyrode's medium supplemented with 25 mM sodium bicarbonate, 22 mM lactate, 1 mM pyruvate, 6 mg ml$^{-1}$ fatty acid–free BSA containing 100 U ml$^{-1}$ penicillin and 100 µg ml$^{-1}$ streptomycin)[65].

For live cell imaging, spermatozoa were then incubated for 30 min with 200 nM mitotracker red FM® (Molecular Probes Inc., Eugene, USA) in fertilization medium[19]. The mitotracker stained spermatozoa were then washed three times in fertilization medium by centrifuging at 100×*g* for 5 min and used for incubation with the oviduct-on-a-chip.

**Live cell imaging.** After 14 days of culture, the oviduct-on-a-chip platform was incubated with Mitotracker red labeled sperm, and stained with Hoechst 33342 (5 µg ml$^{-1}$) in the chip culture medium for 30 min. Live cell imaging was performed by laser scanning confocal microscopy using a ×20 NA 1.25 objective.

**Oocyte collection and in vitro maturation (IVM).** Bovine ovaries were collected from a local abattoir and transported to the laboratory within 2 h. The ovaries were washed in physiological saline (0.9% w/v NaCl) and held in physiological saline containing 100 U ml$^{-1}$ penicillin and 100 µg ml$^{-1}$ streptomycin at 30 °C. Follicular fluid and cumulus-oocyte complexes (COCs) were aspirated from follicles with a diameter of 2 to 8 mm and collected into 50 ml conical tubes using a 19-gauge needle and a vacuum pump[65]. COCs with a minimum of three layers of intact cumulus cells were selected and washed first in HEPES-buffered M199 (Gibco BRL, Paisley, U.K.) before being washed and cultured in maturation medium (M199 supplemented with 0.02 IU ml$^{-1}$ follicle-stimulating hormone [Sioux Biochemical Inc., Sioux Center, IA], 10% FCS, 100 U ml$^{-1}$ penicillin and 100 µg ml$^{-1}$ streptomycin) in four-well culture plates (Nunc A/S, Roskilde, Denmark). Groups of 50 COCs in 500 µl maturation medium were incubated in a humidified atmosphere of 5% CO$_2$-in-air for 24 h at 38.5 °C.

**In vitro fertilization, culture, and embryo collection.** At day 11 of BOEC culture, the apical medium was replaced by fertilization medium (supplemented with 10 µg ml$^{-1}$ heparin, 20 µM d-penicillamine, 10 µM hypotaurine, and 1 µM epinephrine) and a total of 27 in vitro matured COCs were added to the apical compartment of each chip (*n* = 20 devices); sperm was then added at a final concentration of 1 × 10$^6$ sperm cells ml$^{-1}$. The chips were maintained under perfusion (5 µl h$^{-1}$ flow: fertilization medium in the apical compartment and chip culture medium in the basolateral compartment). After 20–22 h of co-incubation under a humidified atmosphere of 5% CO$_2$ and 7% O$_2$ at 38.5 °C, the presumptive zygotes (on chip zygotes) were collected from the apical compartment, cumulus cells were removed by pipetting, and the zygotes were either fixed in 4% paraformaldehyde for 30 min at room temperature (*n* = 30) or frozen for RNA extraction (*n* = 10). Likewise, for conventional IVF, in vitro matured COCs were distributed into groups of 35–50 in four-well culture plates (Nunc A/S, Roskilde, Denmark) with 500 µl of fertilization medium supplemented with 10 µg ml$^{-1}$ heparin, 20 µM d-penicillamine, 10 µM hypotaurine, and 1 µM epinephrine, and spermatozoa were added at a final concentration of 1 × 10$^6$ sperm cells ml$^{-1}$ (normal IVF). Note that matured COCs were randomly distributed between the VT and CH groups. After 20–22 h of co-incubation under a humidified atmosphere containing 5% CO$_2$ and 20% O$_2$ at 38.5 °C, cumulus cells were removed by pipetting and the presumptive zygotes (in vitro zygotes) were fixed (*n* = 30) and/or frozen (*n* = 10) as described above. Remaining zygotes were placed back into the apical compartment of the microfluidic chips and cultured under 5 µl h$^{-1}$ flow, with synthetic oviductal

medium (SOF medium) in the apical and chip culture medium in the basolateral compartment in a humidified atmosphere of 5% $CO_2$ and 7% $O_2$ at 38.5 °C. At days 2 and 4, embryos were scored respectively for cleavage or development to the 8–16 cells stage. For conventional IVP, following denudation presumptive zygotes were distributed in groups of 35–50 in four-well culture plates with 500 μl of SOF medium. The embryos were cultured in a humidified atmosphere of 5% $CO_2$ and 5% $O_2$ at 38.5 °C. At day 4 post-fertilization all 8–16 cells embryos were counted.

**Animal preparation for embryo collection**. Eight Simmental heifers aged between 15 and 20 months and weighing between 380 and 500 kg were used in this study. All experimental animals were handled according to German animal experimentation laws and kept under identical farm conditions within the same herd. Permission was given by the Landesamt für Natur, Umwelt und Verbraucherschutz Nordrhein-Westfalen with reference number 84–02.04.2015.A083 on the 18th of May 2016.

Pre-synchronization of animals was performed by i.m. administration of 500 μg Cloprostenol (a PGF2α analogue, Estrumate ®; Essex Tierarznei, Munich, Germany) twice with an 11 days interval. Two days after each of PGF2α treatment, animals received 20 μg of GnRH (Receptal®; Intervet, Boxmeer, the Netherlands) by i.m. administration. Twelve days after the last GnRH injection, heifers received the first of eight consecutive FSH-injections over 4 days in decreasing doses (in total 400 mg of FSH equivalent according to body weight; Stimufol®, University of Liege, Belgium). Two PGF2α treatments were performed 60 and 72 h after the initial FSH injection. The first of a total of three artificial inseminations within a 12-h interval was performed 48 h after the first PGF2α injection. Finally, 60 h after the first PGF2α application, coincident with the second insemination, ovulation was induced by administration of 10 mg of GnRH.

**Collection of in vivo zygote stage embryos**. Zygotes were collected 19–23.5 h after expected ovulation. For flushing, after restraining the cow, inducing epidural anesthesia with 5 ml of a 2% lidocaine solution (Xylanest ®, Richter Pharma, Wels, Austria) and disinfecting the vulva (Octenisept, Schülke/Mayer, Vienna, Austria), a trocar set consisting of a metal tube (12.5 mm × 52 cm, Storz, Vienna, Austria) and an atraumatic mandrin was placed caudodorsal to the fornix vagina. The mandrin was replaced by a sharp trocar, and the trocar set was inserted through the vaginal wall into the peritoneal cavity. The trocar was replaced by a shaft bearing the endoscope (5.5 mm forward Hopkins endoscope; Storz) and the transfer system. The site was illuminated using a fiberoptic cold light (250 W, Storz) and visualized with a camera (Telecam PAL-Endovision, Storz) connected to a monitor. The flushing system consisted of a 20-ml syringe connected to a perfusor tube (No. 08272514; Braun, Melsungen, Germany) and a metal tube (14 cm × 2.5 mm) with numerous lateral holes covered by a silicone tube. After the metal tube had been inserted via the infundibulum into the ampulla, careful management of the flushing pressure allowed the balanced adjustment of tubal sealing to avoid medium reflux. Oviducts were flushed with 50 ml flushing medium (phosphate-buffered saline supplemented with 1% fetal calf serum). Flushing medium (50 ml) was forced through the uterotubal junction into the uterine horn and from there was collected via a uterus flushing catheter (CH15, Wörrlein, Ansbach, Germany) into an embryo filter (Emcon filter, No. 04135; Immuno Systems Inc., Spring Valley, WI, USA)[66].

**Immune fluorescence for global methylation**. Immunofluorescent staining for 5-methylcytosine (5mC) was performed in zygotes at different pronuclear stages. Fixed zygotes were permeabilized by incubation for 30 min in 1% Triton-X100 in PBS, followed by denaturation with 3 M HCl for 30 min, which was then neutralized using 100 mM Tris-HCl buffer (pH 8.5) for 15 min. Non-specific binding was blocked by incubating the permeabilized zygotes for 1 h in PBS containing 5% normal goat serum (NGS). The zygotes were then incubated overnight at 4ºC with a mouse anti-5mC primary antibody (1:100 dilution with PBS + 5% NGS + 0.1% Triton-X100; Eurogentec, BI-MECY-0100). Next, the zygotes were washed three times in PBS-polyvinylpyrrolidone (PVP, 3 mg ml$^{-1}$; 10 min each) and incubated with an Alexa 488 conjugated goat anti-mouse antibody (1:100, Santa Cruz Biotechnology, Santa Cruz, CA) for 1 h. Zygotes were then incubated with propidium iodide (PI; 25 μg ml$^{-1}$) for 30 min to counterstain the DNA. Negative controls were produced by omitting incubation with the primary antibody. Analysis was performed by laser scanning confocal microscopy with a ×40 NA 1.25 objective. Z-stacks of 1 μm of both pronuclei were obtained. Images were analyzed by evaluating the fluorescence intensity for 5mC and DNA in each pronucleus using ImageJ software (National Institutes of Health, Bethesda, MD, USA). After maximum projection reconstruction of Z-stacks, the fluorescence intensity of each channel was measured by manually outlining each pronucleus and adjusted for cytoplasmic background. The average intensity of 5 mC fluorescence was then adjusted by dividing by PI fluorescence to normalize the 5 mC intensity for DNA content.

**RNA extraction**. Cells were collected from the chips by perfusing with 200 μl of kit lysis buffer for 2 min (RNEasy Micro RNA extraction kit; Qiagen GmbH, Hilden, Germany). Total RNA was isolated from single zygotes, or from the cells collected from the devices using the RNEasy Micro RNA extraction kit; (Qiagen GmbH,

Hilden, Germany), and treated with RNAse-free DNAse I (Qiagen GmbH, Hilden, Germany) to remove genomic DNA, following the manufacturer's instructions.

**Cel-seq II primer design**. The reverse-transcription primer was designed with an anchored polyT, a 6 bp unique barcode, a 6 bp UMI (unique molecular identifier), the 5′ Illumina adapter and a T7 promoter. The barcodes were designed such that each pair was different by at least two nucleotides, so that a single sequencing error would not produce the wrong barcode (adapted from Hashimshony et al.[67]).

**Linear mRNA amplification**. RNA extracted from single zygotes and from BOECs was precipitated with isopropanol and the pellet was used for the reverse-transcription (RT) reaction. RT was performed with 5 ng of primer per reaction. A total of 0.2 μl of the primer mixed with 1 μl of water or 1 μl of a 1:1,000,000 dilution of the ERCC spike-in kit (a total of 1.2 μl) was added directly to the Eppendorf tube in which the RNA was precipitated, and incubated at 65 °C for 5 min (with the lid of the thermal cycler heated to 65 °C). The sample was spun to the bottom of the tube mid incubation. After the second-strand synthesis, samples were pooled and cleaned on a single column before proceeding to the IVT (Ambion AM1334) reaction for 13 h. The solution was treated with EXO-SAP to remove the primers and the RNA was fragmented (one-fifth volume of 200 mM Tris-acetate [pH 8.1], 500 mM KOAc, 150 mM MgOAc added) for 3 min at 94 °C. The reaction was stopped by placing the sample on ice and adding one-tenth volume of 0.5 M EDTA, followed by RNA cleanup. The RNA quality and yield were analyzed using a Bioanalyzer (Agilent).

**Library construction and Cel-seq II**. RT reaction was performed using Super-Script II, following the manufacturer's protocol (Invitrogen). A total of 14 cycles of PCR was performed using Phusion® High-Fidelity PCR Master Mix with HF Buffer (NEB, MA, USA) and an elongation time of 30 s. PCR products were cleaned twice with AMPure XP beads (Beckman Coulter, Woerden, Netherlands). Libraries were sequenced on the Illumina Nextseq500 platform; a high output paired end run of 2 × 75 bp was performed.

**Cel-seq II data analysis**. Differentially expressed genes were identified using the Deseq2 (v1.10.1) package[68]. Genes with low counts (whose sum of all counts across samples included in the analysis was <10) were removed. The $p$-value was determined by Wald statistics. An adjusted $p$-value to correct for multiple testing was calculated using the Benjamini–Hochberg method. Differentially expressed genes (DEGs) were filtered by fold change (lfc Threshold = 1) and a false discovery rate (FDR) <1% (alpha = 0.1). Biological functions of differentially regulated gene sets were identified using ToppGene Suite tool ToppFun (default setting: FDR correction, $p$-value cutoff of 0.05 and gene limit set of $1 \le n \le 2000$[69]).

**Functional GO clustering**. The Cytoscape 3.5.1 plugin ClueGO[70] was used to functionally group the upregulated and downregulated genes by GO terms "biological processes" and "cellular components" using the *Bos taurus* genome. The evidence was set to "Inferred by Curator (IC)," and the statistical test was set to a right-sided hypergeometrical test with a Bonferroni (step down) and a κ score of 0.7–0.9. The GO term restriction levels were set to 3–8, with a minimum of three genes or 5% genes in each GO term, and the function "GO Term fusion" was selected.

**Data analysis**. The data were analyzed using IBM SPSS Statistics (version 24). A Shapiro–Wilk test was performed, and all data proved to be normally distributed. Mean and standard deviations are provided in graphs; differences between groups were examined by ANOVA followed by a post-hoc Tukey test with a confidence interval of 95%.

## Data availability

The authors declare that all data supporting the findings of this study are available within the article and its Supplementary Information Files or from the corresponding author upon reasonable request. Total RPM counts of the CellSeq II RNA-sequencing of BOECs and zygotes have been deposited in FIGSHARE database under https://doi.org/10.6084/m9.figshare.7157150 (https://figshare.com/articles/Oviduct-on-a-chip_RNAseq_data_for_cells_and_zygotes/7157150).

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

## Acknowledgements

We would like to thank Dr. Eugene Y. Kathruka for his help with live imaging of the chips; Nuno F. Pinheiro for his help in modeling chips for flow rate and shear stress calculations; Christine Oei for detecting the progesterone levels in the collected fluids in the cell-free chips perfusion experiment, and Juliette Delahaye for her help in designing the microfluidic devices.

## Author contributions

M.A.M.M.F. designed and performed all experiments and wrote the manuscript. S.L. G. and H.S.R. designed and produced the microfluidics devices and reviewed the manuscript. H.T.A.v.T. performed in vitro fertilization experiments and reviewed the manuscript. M.H. and U.B. performed in vivo collection of embryos and reviewed the manuscript. D.H. and M.M. performed RNA-seq data analysis. B.M.G., H.H.W.H., T.A.E.S., P.L.A.M.V., supervised M.A.M.M.F., designed experiments, and reviewed the manuscript. B.M.G. performed the cell-free chips perfusion experiments and was main supervisor of this research project.

## Additional information

**Competing interests:** The authors declare no competing interests.

