## [Peer Review File · Nature Communications]

Reviewers' comments:

Reviewer #1 (Remarks to the Author):

The impact of the need for treatment for infertility in humans is being addressed across the world. None the less, the potential for long term consequence is not currently the central focus primarily as those born from these procedures are only now in their late thirties and the relationship of development and adult health and disease has not yet emerged. This study focuses on the importance of the physical environmental conditions in which IVF and other post ART procedures take place and the influence they may have on embryo viability as measured by readouts from the epigenome. This is an interesting and cutting edge technological invention for creating a more 'in vivo' environment in which early maturation and post fertilisation culture may take place. The use of microfluidic chambers with co-culture of oviduct epithelia cells has potential for significant improvement of these DoAHD challenges.

This type of system has unprecedented value for the long term, tightly controlled maintenance of oocytes and embryos in a cost effect and precision medical context for the treatment of infertility.

Here the authors use bovine as a model system, which gives a better approximation to a number of reproductive and physiological features more similar to humans than the more usual mouse model system.

I found this manuscript very well written and extremely interesting to read.

The author's system focuses around the zygote and measures DNA methylation by immunofluorescence and RNA profiling by RNA-seq to compare outcomes between the in vitro and in vivo groups. The proxy for different periods of oocyte development through addition of key hormones clearly has an effect and there are subtle response as measured by RNA-seq.

It is not immediately clear what is novel about the current study. This collection, some by these authors, cover the salient points of the method.

1. Ferraz, M. A. M. M. et al. Improved bovine embryo production in an oviduct-on-a-chip system: Prevention of poly-spermic fertilization and parthenogenic activation. *Lab Chip* (2017). doi:10.1039/C6LC01566B
2. Ferraz, M. A. M. M., Henning, H. H. W., Stout, T. A. E., Vos, P. L. A. M. & Gadella, B. M. Designing 3-Dimensional In Vitro Oviduct Culture Systems to Study Mammalian Fertilization and Embryo Production. *Ann. Biomed. Eng.* (2016). doi:10.1007/s10439-016-1760-x
3. Xiao, S. et al. A microfluidic culture model of the human reproductive tract and 28-day menstrual cycle. *Nat. Commun.* 8, 14584 (2017).

4. Lai, D., Takayama, S. & Smith, G. D. Recent microfluidic devices for studying gamete and embryo biomechanics. *J. Biomech.* 48, 1671–1678 (2015).

The authors refer to the complete reprogramming, which takes place in the oviduct. To those in the reprogramming field this would not be correct and indeed complete reprogramming only occurs in germ cells when genomic imprints are restored in a parent of origin manner. I am assuming that this is a reference to the second reprogramming during preimplantation when DNA methylation is modulated again, significantly while retaining imprinted instructions.

I was not entirely clear why the authors elected to investigate RNA-seq profiles in bovine zygotes when they are only measuring the transcriptome of the oocyte as EGA, according to Graf et al (2014) *Reproduction*, shows a minor burst at the 4-cell stage and the major burst at 8-16 cell stage. As such, what is the biological significance of these changes, which are reported in zygotes? Wouldn't the largest change occur at the blastocyst stage after EGA?

In Fig 5 the authors profile a PCA plot where there are two distinct clusters. These clusters overlap with respect to groups and as such suggest no clear segregation of any treatment group. Can the authors exclude that this isn't a function of batch effects?

One disadvantage of non-mouse models is the limitations of non-genetic uniformity. In this respect, what degree of variation might be expected by genetic variation alone and how is that controlled for in the various experimental groups?

The extensive GO evaluation is difficult to extract specific information from. As a control, one would expect that ER genes would be expressed in the high estrogen environment. Was this the case?

The authors report that DNA methylation as measured by immunofluorescence is similar between the on-Chip group and the in vivo group and much lower than that of the in vitro group. Here the loss of DNA methylation is attributed to activities associated to 'de-methylation' or loss of methylation mechanisms such as the oxidation by the TET family of proteins. Have the authors considered that there is a much wider group of factors associated with loss of DNA methylation that are connected by virtue of the reduced addition of the methyl group by de novo MTase activity.

In line 388 the authors refer to the use of gene editing for either the gametes or embryos. I can see this for the cell layers but not the embryos. What do you mean by this statement?

Minor points

1. In line 607 I believe you mean immunofluorescence.
2. In line 619 the authors refer to negative controls. Where are these shown?

Reviewer #2 (Remarks to the Author):

In the manuscript entitled “An oviduct-on-a-chip provides an enhanced in vitro environment for zygote (epi)genome reprogramming”, authors used the microfluidic technology to culture the bovine oviduct epithelial cells to establish an oviduct chip and then they fertilized the oocytes and cultured the embryos till blastocyst stage and compare the reproductive outcomes with embryos from traditional petri-dish culture and from in vivo. It was found that the oviduct chip resulted in more physiological embryo epigenetic reprogramming and gene transcriptome. Authors concluded that the oviduct-on-a-chip could improve the quality and epigenetic integrity of IVF embryos, which could benefit the later implantation, placentation, fetal development, and even postnatal development. This is a very novel study and the results generated are promising and indicate that the oviduct chip has great potential to help produce better quality of embryos when animals and humans are using assisted reproductive technology for reproduction.

However, more detailed and convincing data are also necessary to demonstrate this central hypothesis. The following comments and suggestions listed below could significantly promote the manuscript.

1. In the “Introduction”, it is highly recommended that authors include a summarized description of the essential roles and known mechanisms of in vivo oviductal environment on ovum fertilization and preimplantation embryo development and epigenetic reprogramming, and how the previous studies demonstrated that the traditional IVP on flat plastics cannot completely recapitulate the in vivo regulation. These are the focuses of the current study and addressing these will make audiences easy to follow the manuscript.

2. It is a good idea that authors tried to use the oviduct chip to mimic the in vivo physiological environment, however, the results generated cannot demonstrate whether the beneficial effects are caused by the microfluidic culture technique or only by the use of oviductal cell-embryos co-culture. More evidences are necessary to differentiate this such as the 3D oviduct and embryo co-culture in non-microfluidic culture environment.

3. With respect to the two independent compartments, authors used the bottom part to provide a dynamic hormonal environment by supplementation of estrogen and progesterone. Is it possible that the hormones in the bottom compartment contaminate the top part? The TEER assay is a good experimental design to address this but cannot fully answer this question. Besides, the cell confluence and cell layer permeability are also different at different phases upon hormone treatment. Using the conditional media (with only oviduct cell culture without the embryo) to measure the hormonal concentrations may figure this out.

4. Authors indicated that the pillars were used to trap the oocytes and embryos. We know that the embryos will also transport from the fallopian tube to uterus during preimplantation embryo development period. Is the missing of embryo transport will affect the embryo epigenetic reprogramming?

5. The BOESc cultured in the oviduct chip enables the positive expression of some genes critical for sperm-oviduct adhesion, COC-oviduct interaction, and embryo development. However, it is difficult to conclude that this is caused by the microfluidic culture without comparing to the BOESc culture in the non-microfluidic culture environment. If the microfluidic culture is the case, what is the specific mechanism, such as dynamic flow, dynamic hormone treatment, or oviduct-embryo contact?

6. Line 232-236: More evidence and quantification data are necessary to conclude that the compromised success of in vitro embryo development outcomes is caused by the pillar trapping and lost along the dynamic perfusion and to exclude the factor of the culture environment of oviduct chip.

7. Authors used the fluorescent 5mC staining to indicate the global methylation level and find that the embryos from oviduct chip is closer to the in vivo embryos than that from the traditional in vitro culture. This is promising preliminary data. It will be great if authors could produce more specific gene DNA methylation and then compare the patterns among three groups, such as the whole genomic methylome study or the DNA methylation and gene expression levels of imprinted genes.

Reviewer #3 (Remarks to the Author):

The manuscript describes the culture of bovine oviduct epithelial cells (BOECs) in an organ chip. The chip is divided into two departments, separated by a porous membrane seeded with the BOECs. The basolateral side is used to flush hormones into the system, while the cells are cultured on the apical side in culture medium without hormones.

Next, this chip is used to support in vitro fertilisation and embryo development.

The first part of development of the oviduct-on-a-chip is very convincing. There are four minor recommendations I would like to add (not obligatory for the acceptance of paper in my view):

- 1) It would be nice to include a graphical visualisation of polarisation of BOECs cells, similar to this publication: Yi Duan et al (2010) Shear stress-induced changes of membrane transporter localization and expression in mouse proximal tubule cells
- 2) Include data on cilia beating frequency.
- 3) Could you add information about the physiological range of TEER in (bovine) ovaries?
- 4) Could you give more information on the decision of culture time
 - a. Why was the chip cultured in static for 4 days was the nutrient supply sufficient during this period?
 - b. What is the reason to keep the luteal phase for 7 days before the pre-ovulatory phase?

It is known that PDMS adsorbs hydrophobic substances. The logP value of estrogen and progesterone are rather high. Therefore, an adsorbance into the PDMS can be expected and the final concentration of hormones reaching the cells can be drastically reduced. Has this been analysed? Additionally, there might be some release of (adsorbed) hormones, once the concentration of progesterone is reduced. This might lead to an increased concentration of progesterone in the pre-ovulatory phase.

The data show a difference between the different hormone applications, however it would be helpful to know, how much of the hormones effectively reached the cells.

Major critics are concerning the in chip fertilisation:

The cleavage and blastocyst formation was significantly lower than in in vitro embryo production.

1) What is the benefit of fertilisation with flow (what is the reason to keep the flow after a confluent layer of BOECs formation)?

a. How does this correspond to the physiological flow rate?

b. Would a pulsatile flow or an on/off regimen increase the yield?

c. Why is it not sufficient to perfuse the basolateral side of the chip and avoid any shear stress on the apical side?

d. Alternatively, reduce the flow on the apical side?

e. Is it necessary to use a chip or would it be sufficient to plate BOECs on a petri dish?

2) The design of the columns is obviously not optimal for the oocytes. The oocytes are trapped and the shear stress is too high. A different chip design would probably increase the cleavage and blastocyst formation. Why did you stick to this design?

Reviewer #1 (Remarks to the Author):

The impact of the need for treatment for infertility in humans is being addressed across the world. None the less, the potential for long term consequence is not currently the central focus primarily as those born from these procedures are only now in their late thirties and the relationship of development and adult health and disease has not yet emerged. This study focuses on the importance of the physical environmental conditions in which IVF and other post ART procedures take place and the influence they may have on embryo viability as measured by readouts from the epigenome. This is an interesting and cutting edge technological invention for creating a more 'in vivo' environment in which early maturation and post fertilisation culture may take place. The use of microfluidic chambers with co-culture of oviduct epithelia cells has potential for significant improvement of these DoAHD challenges.

This type of system has unprecedented value for the long term, tightly controlled maintenance of oocytes and embryos in a cost effective and precision medical context for the treatment of infertility.

Here the authors use bovine as a model system, which gives a better approximation to a number of reproductive and physiological features more similar to humans than the more usual mouse model system.

I found this manuscript very well written and extremely interesting to read.

The author's system focuses around the zygote and measures DNA methylation by immunofluorescence and RNA profiling by RNA-seq to compare outcomes between the in vitro and in vivo groups. The proxy for different periods of oocyte development through addition of key hormones clearly has an effect and there are subtle response as measured by RNA-seq.

It is not immediately clear what is novel about the current study. This collection, some by these authors, cover the salient points of the method.

1. Ferraz, M. A. M. M. et al. Improved bovine embryo production in an oviduct-on-a-chip system: Prevention of poly-spermic fertilization and parthenogenic activation. *Lab Chip* (2017). doi:10.1039/C6LC01566B

Our originally published system is improved in a way that we can culture embryos on top and keep the apical compartment (cell compartment) under constant flow (which is necessary to keep cells polarized). The other systems are hampered either with the use of air-liquid interphase (no fluid in the apical compartment to have epithelial cell differentiation) or with static media on top in which the cells lost their differentiation (become flat and lose ciliation; see ref 24). The maintenance of epithelial cell differentiation under 5 uL/h perfusion conditions is the novelty of the current work.

2. Ferraz, M. A. M. M., Henning, H. H. W., Stout, T. A. E., Vos, P. L. A. M. & Gadella, B. M. Designing 3-Dimensional In Vitro Oviduct Culture Systems to Study Mammalian Fertilization and Embryo Production. *Ann. Biomed. Eng.* (2016). doi:10.1007/s10439-016-1760-x

That reference is a review in which features of models are described in a more generalized fashion. The manuscript mentioned under point 1 describes the results we obtained from a first oviduct on a chip model which was hampered by toxicity issues (see Ferraz et al.,

ENVIRONMENTAL SCIENCE & TECHNOLOGY

LETTERS Volume: 5 Issue: 2 Pages: 80-85 Published: FEB 2018). Now included as ref. 22.

The current manuscript describes the use of non-toxic plastics, the use of perfusion keeps epithelial cell differentiation (see answer to point 1) and the use of hormonal switch of the basolateral perfusion fluid to induce activation of the BOEC.

3. Xiao, S. et al. A microfluidic culture model of the human reproductive tract and 28-day menstrual cycle. *Nat. Commun.* 8, 14584 (2017).

In that manuscript they perform tissue pieces culture with a concomitant disadvantage: The tissue will not completely cover the membrane and consequently there is no separated apical and basolateral perfusion compartment. For that reason the manipulations performed in our current study at the basolateral compartment of our chip (after establishing a functional confluency and differentiation of our BOEC) cannot be performed on the model of Xiao et al. In addition, by culturing cells and not tissues, we can actually genetic manipulate the cells (like using CRISPr/Cas9), to study effects of knocking out specific genes on the cells and on embryo development; which is not possible when using pieces of tissues. As mentioned in the discussion (lines 444-455)

4. Lai, D., Takayama, S. & Smith, G. D. Recent microfluidic devices for studying gamete and embryo biomechanics. *J. Biomech.* 48, 1671–1678 (2015).

In that review manuscript the authors do not include effects of oviductal cells, but indeed describes the effects of different microfluidics devices on gametes/embryos.

These references were already referred to in our original submission. We strongly believe that our new microfluidic approach is an improvement of all cited models above.

The authors refer to the complete reprogramming, which takes place in the oviduct. To those in the reprogramming field this would not be correct and indeed complete reprogramming only occurs in germ cells when genomic imprints are restored in a parent of origin manner. I am assuming that this is a reference to the second reprogramming during preimplantation when DNA methylation is modulated again, significantly while retaining imprinted instructions. Indeed, we refer to the reprogramming that happens after fertilization. We are sorry for the misunderstanding by using the “complete” term and rephrased accordingly in line 78-79.

I was not entirely clear why the authors elected to investigate RNA –seq profiles in bovine zygotes when they are only measuring the transcriptome of the oocyte as EGA, according to Graf et al (2014) *Reproduction*, shows a minor burst at the 4-cell stage and the major burst at 8-16 cell stage. As such, what is the biological significance of these changes, which are reported in zygotes? Wouldn't the largest change occur at the blastocyst stage after EGA?

For the past few years, a number of publications evaluating DNA methylation has shown that in vitro conditions have a great influence on zygotic methylation. Also, at this stage, there are differential demethylation of the paternal and maternal pronuclei and we believe it is a critical point for the developing embryo. A study of Zongliang Jiang et al., (2015) has shown that in vivo bovine embryos already have differences on gene expression of imprinted genes between the oocyte to the 2 cell stage (such as MEST, PLAGL1, CDKN1C, IGF2R, SGCE, PEG10, PHDA2). Furthermore, Zongliang Jiang et al., (2015'2) had demonstrated that there are four waves of transcriptional changes, between oocyte and 2-cell, between 4- and 8-cell, between 16-cell to early morula, and between compact morula to blastocyst. Combining all these finds, with

the fact that we saw differences on global methylation by IF between in vivo, in vitro and chip zygotes, we decided to perform the RNAseq data using zygotes.

Blastocyst wouldn't be an option in our study, since the embryo leaves the oviduct between the 8-16 cell stage, which can lead to the oviduct not being a suitable environment for later stages of embryo development.

That could also partially explain why the blastocyst rates were lower on the chip when compared to the normal IVF. So, we decided to remove blastocyst rates and include the 8-16 cells rate in the manuscript, for a more accurate stage of which embryos stay in the oviduct. This information is now included in lines 286-289.

In Fig 5 the authors profile a PCA plot where there are two distinct clusters. These clusters overlap with respect to groups and as such suggest no clear segregation of any treatment group. Can the authors exclude that this isn't a function of batch effects?

Indeed we cannot exclude if it is a function of batch, but collecting a higher number of in vivo embryos is difficult. Also, we performed single zygotes RNAseq, and 80% of in vivo zygotes clustered pretty well together, and 100% of vitro zygotes (collected in 3 different replicates) also did cluster together, so we believe that if batch was the case, we would have more disparities inside the same groups of zygotes.

One disadvantage of non-mouse models is the limitations of non-genetic uniformity. In this respect, what degree of variation might be expected by genetic variation alone and how is that controlled for in the various experimental groups?

We agree with this notion and indeed did not have control of the genetic variation for the groups. Also, we did not have access to animal background for in vitro samples. However, we do not expect large differences between materials obtained as it was always collected in a routine way at the same slaughterhouse. And all collected oocytes from different cows were mixed and in vitro matured. After maturation the COCs were randomly divided between the VT and CH groups for in vitro fertilization. As described in lines 631-632.

The extensive GO evaluation is difficult to extract specific information from. As a control, one would expect that ER genes would be expressed in the high estrogen environment. Was this the case?

Indeed the estrogen related receptor alpha (ESRRA) was up regulated in the pre-ovulatory phase as well as an increased expression of progesterone receptor membrane components 1 and 2 (PGRMC1 and PGRMC2) were observed on luteal phase. Information included in text (Lines: 222-223 and 237-239).

The authors report that DNA methylation as measured by immunofluorescence is similar between the on-Chip group and the in vivo group and much lower than that of the in vitro group. Here the loss of DNA methylation is attributed to activities associated to 'de-methylation' or loss of methylation mechanisms such as the oxidation by the TET family of proteins. Have the authors considered that there is a much wider group of factors associated with loss of DNA methylation that are connected by virtue of the reduced addition of the methyl group by de novo MTase activity.

We agree with this and do think DNMTs are important and we actually saw differences on DNMTs expressions in in vivo and in vitro groups. In a more in vivo-like environment the

DNMT3b were the most expressed DNMT, while in vitro DNMT1 is the most abundant. We showed that in vitro-like embryos were more prone to methylation maintenance (DNMT1) while in vivo-like embryos were more prone to de novo methylation (DNMT3b). We changed figure 5 accordingly and included table I in text.

In line 388 the authors refer to the use of gene editing for either the gametes or embryos. I can see this for the cell layers but not the embryos. What do you mean by this statement?

We are sorry for the lack of information provided. There is a new technology for embryo gene editing without microinjection: Genome-editing via Oviductal Nucleic Acids Delivery (GONAD). Using GONAD, it was shown that NAs (e.g., eGFP mRNA or Cas9 mRNA/sgRNAs) can be effectively delivered to pre-implantation embryos within the intact mouse oviduct by a simple electroporation method, and result in the desired genetic modification in the embryos (Takahashi et al 2015). We believe that the same principle could be used by using our oviduct-on-a-chip.

Text was corrected in lines 429-430 to include this information.

Minor points

1. In line 607 I believe you mean immunofluorescence.

Corrected in text.

2. In line 619 the authors refer to negative controls. Where are these shown?

We have not included these, the negative controls showed no staining and this information has now been included in the legend of Figure 4.

Reviewer #2 (Remarks to the Author):

In the manuscript entitled “An oviduct-on-a-chip provides an enhanced in vitro environment for zygote (epi)genome reprogramming”, authors used the microfluidic technology to culture the bovine oviduct epithelial cells to establish an oviduct chip and then they fertilized the oocytes and cultured the embryos till blastocyst stage and compare the reproductive outcomes with embryos from traditional petri-dish culture and from in vivo. It was found that the oviduct chip resulted in more physiological embryo epigenetic reprogramming and gene transcriptome. Authors concluded that the oviduct-on-a-chip could improve the quality and epigenetic integrity of IVF embryos, which could benefit the later implantation, placentation, fetal development, and even postnatal development. This is a very novel study and the results generated are promising and indicate that the oviduct chip has great potential to help produce better quality of embryos when animals and humans are using assisted reproductive technology for reproduction. However, more detailed and convincing data are also necessary to demonstrate this central hypothesis. The following comments and suggestions listed below could significantly promote the manuscript.

1. In the “Introduction”, it is highly recommended that authors include a summarized description of the essential roles and known mechanisms of in vivo oviductal environment on ovum fertilization and preimplantation embryo development and epigenetic reprogramming, and how

the previous studies demonstrated that the traditional IVP on flat plastics cannot completely recapitulate the in vivo regulation. These are the focuses of the current study and addressing these will make audiences easy to follow the manuscript.

This was already included in lines 49-62 of the introduction and pertinent literature for more details was also included in this paragraph. We didn't elaborate further on this because of the lack of space in the manuscript.

2. It is a good idea that authors tried to use the oviduct chip to mimic the in vivo physiological environment, however, the results generated cannot demonstrate whether the beneficial effects are caused by the microfluidic culture technique or only by the use of oviductal cell-embryos co-culture. More evidences are necessary to differentiate this such as the 3D oviduct and embryo co-culture in non-microfluidic culture environment.

As noted in a reply to reviewer 1 (on manuscript 1 that the reviewer mentioned): When non-microfluidic oviductal cultures are used the polarized BOEC will start to become flat and lost their differentiated state (cf. ref. 23-25). This is a stressful changing environment for the cells. Therefore, we don't believe that having a static culture is a comparable control. Microfluidics during embryo culture is necessary to keep the cells in this polarized state, that's why we didn't have a control without constant perfusion of epithelial cells. Note also that a higher perfusion speed (30 $\mu\text{L/h}$) resulted in blebbing and deterioration of the BOEC cells. This information is now included in the text lines 130-133 and in the new suppl. fig. 1.

3. With respect to the two independent compartments, authors used the bottom part to provide a dynamic hormonal environment by supplementation of estrogen and progesterone. Is it possible that the hormones in the bottom compartment contaminate the top part? The TEER assay is a good experimental design to address this but cannot fully answer this question. Besides, the cell confluence and cell layer permeability are also different at different phases upon hormone treatment. Using the conditional media (with only oviduct cell culture without the embryo) to measure the hormonal concentrations may figure this out.

We did combine TEER measurements with a paracellular assay, in which dyes of 0.4 and 4 μm were used, and the monolayer of BOEC were impermeability to these dyes. Therefore, only by passing through the monolayer of BOEC hormones can reach the apical compartment of the device. The design of this was to allow passing through of steroids through these cells as is the case in the in vivo situation.

4. Authors indicated that the pillars were used to trap the oocytes and embryos. We know that the embryos will also transport from the fallopian tube to uterus during preimplantation embryo development period. Is the missing of embryo transport will affect the embryo epigenetic reprogramming?

We believe that the different regions of the oviduct and the transport are also important for embryo quality, not necessarily epigenetic reprogramming only. Nevertheless, a major problem on removing the pillars is that the epithelial cells do need the constant flow on top of it to keep them polarized. With the same flow embryos will be washed away from the chip. Considering this we decided that the trapping pillars were the best option to have the BOEC cells stimulated to keep their polarization and to produce embryos inside the chip. We are also working with Dr. le Gac's group on developing new systems in which we can have both ampulla and isthmus cells on it. Furthermore, we are still designing models in which embryos can move more freely inside

the chip. This is not discussed in lines 109-111.

5. The BOECs cultured in the oviduct chip enables the positive expression of some genes critical for sperm-oviduct adhesion, COC-oviduct interaction, and embryo development. However, it is difficult to conclude that this is caused by the microfluidic culture without comparing to the BOESc culture in the non-microfluidic culture environment. If the microfluidic culture is the case, what is the specific mechanism, such as dynamic flow, dynamic hormone treatment, or oviduct-embryo contact?

We do not have an answer to this question: Mechano-sensory receptor signaling pathways are described in literature to be involved in cell specific responses. In our system the apical flow rate of medium was important to maintain the differentiation of the BOEC. Flow rate of 0 $\mu\text{L}/\text{h}$ resulted in flattening and dedifferentiation (cf. refs 23-25) while 30 $\mu\text{L}/\text{h}$ resulted in cell blebbing and cell deterioration (see newly included suppl. fig. 1). The 5 $\mu\text{L}/\text{h}$ was suitable for maintaining the cell differentiation of the BOEC. The hormone treatment (at the basolateral compartment only) was to activate the differentiated BOEC.

There are other ways to keep BOECs polarized and with a morphology and function similar to the *in vivo* tissue:

1. By using membrane inserts and air-liquid interface
2. By producing organoids

Using air-liquid interface is not compatible with both BOEC polarization and embryo culture, because when producing embryos we will have static medium on top and the cells will lose their polarized state. There is a recent publication in which they produced embryos using polarized BOECs in membrane inserts, and low volume of media on top of cells (van der Weijden et al., 2017), in which they tested 41 genes and compared BOEC embryos with regular IVP embryos and couldn't find any difference on gene expression.

Organoids models of oviducts were already successfully created (Kessler et al., 2015), but using organoids to IVF and embryo culture is not a viable option, since the cells are polarized to the inside of the organoid, making it not possible to perform IVF and IVC inside of it.

Anyway, the main point is that we can have BOECs in an *in vivo*-like morphology and function by using different systems, but microfluidics is the one system that allows us to perform IVF and IVC, without compromising the BOECs polarization.

6. Line 232-236: More evidence and quantification data are necessary to conclude that the compromised success of *in vitro* embryo development outcomes is caused by the pillar trapping and lost along the dynamic perfusion and to exclude the factor of the culture environment of oviduct chip.

We agree with this point raised, that's why we mentioned that the loss of embryos can be a cause, together with shears stress. We also produced embryos in a non-hormone stimulation environment. Future experiments will compare different hormonal stimulations which may improve fertilization and embryo development rates. Note that we cultured embryos until the blastocyst stage, but the embryo leaves the bovine oviduct at 8-16 cell stages. The oviduct environment may not be ideal for further development into blastocysts. We therefore changed the

text (results, discussion and materials and methods), by replacing the rates of development into blastocyst stage into 8-16 cells rates.

7. Authors used the fluorescent 5mC staining to indicate the global methylation level and find that the embryos from oviduct chip is closer to the in vivo embryos than that from the traditional in vitro culture. This is promising preliminary data. It will be great if authors could produce more specific gene DNA methylation and then compare the patterns among three groups, such as the whole genomic methylome study or the DNA methylation and gene expression levels of imprinted genes.

Indeed whole genome methylome would be the best option which is planned for future experiments. We would need enough material for whole methylome and would need much higher number of embryos which need to be pooled. The problem is to get such high numbers of in vivo embryos. Therefore, we decided to complement the 5mC staining with the single embryo RNAseq data. We believe that the immunofluorescence together with the single embryo RNAseq was enough to this manuscript that was mostly focused primarily on the development of the microfluidics system for BOECs culture and its use for in vitro embryo production.

Reviewer #3 (Remarks to the Author):

The manuscript describes the culture of bovine oviduct epithelial cells (BOECs) in an organ chip. The chip is divided into two departments, separated by a porous membrane seeded with the BOECs. The basolateral side is used to flush hormones into the system, while the cells are cultured on the apical side in culture medium without hormones.

Next, this chip is used to support in vitro fertilisation and embryo development.

The first part of development of the oviduct-on-a-chip is very convincing. There are four minor recommendations I would like to add (not obligatory for the acceptance of paper in my view):

1) It would be nice to include a graphical visualisation of polarisation of BOECs cells, similar to this publication: Yi Duan et al (2010) Shear stress-induced changes of membrane transporter localization and expression in mouse proximal tubule cells

We are not sure which the type of information is requested here. We are confident that the way we present our data (immune fluorescent pictures plus cell height and ciliation graphs) is clear and informative enough. However, we are willing to adapt and improve data presentation with more detailed instructions.

2) Include data on cilia beating frequency.

We didn't have the equipment to monitor cilia beating frequency bright field images, also the porous membrane is not transparent, therefore the light doesn't pass through it and the only way of checking for cilia beating would be to perform live imaging with fluorescence dyes. That's why we didn't include it in the manuscript. We tried to label the cilia with the fluorescence dye SiR-Tubulin and perform live imaging to monitor cilia beating rate. But, by accumulating the dye, the cilia stopped their movements and we were not able to keep track of cilia beating.

3) Could you add information about the physiological range of TEER in (bovine) ovaries?

We do not have information about physiological range of TEER in bovine oviducts. It is very difficult to perform such experiments in vivo (and from an animal ethical commission point of view such experiments are difficult to get permission for). We should note (a) that ultrastructural data from the oviduct convincingly showed that the oviduct is lined with a confluent monolayer of epithelial cells (for instance see ref 24) and (b) that we used a combination of TEER measurements with the paracellular flux assay simply to prove that we had a confluent and tight monolayer in culture of BOECs. This is in our view one of the absolute requirements of an in vitro approach to mimic the in vivo situation. The way we detected cell confluency and polarity was described before (see Chen et al., 2015).

4) Could you give more information on the decision of culture time

a. Why was the chip cultured in static for 4 days was the nutrient supply sufficient during this period?

We cultured it static for 4 days, because it is the time the cells need to attach and confluence, if we were perfusing during this period, we would wash cells away and not have a confluent monolayer. Nutrient supply was enough, because media was being changed manually from the basolateral compartment 2 times a day. This information is described in lines 489-495.

b. What is the reason to keep the luteal phase for 7 days before the pre-ovulatory phase?

In total we had an 11 days luteal phase before the pre-ovulatory phase (4 days static and 7 days perfusing). During the average 18-20 days of the cow's estrous cycle, 11 days it is under high progesterone influence (luteal phase) and between 3-4 days it is under high estrogen and low progesterone stimulation, that is the rationality for the hormonal stimulation in this study. The remaining days of the cycle is a progressive increase/decrease of both hormones that we did not mimic in the present experiment.

It is known that PDMS adsorbs hydrophobic substances. The logP value of estrogen and progesterone are rather high. Therefore, an adsorbance into the PDMS can be expected and the final concentration of hormones reaching the cells can be drastically reduced. Has this been analysed? Additionally, there might be some release of (adsorbed) hormones, once the concentration of progesterone is reduced. This might lead to an increased concentration of progesterone in the pre-ovulatory phase.

The data show a difference between the different hormone applications, however it would be helpful to know, how much of the hormones effectively reached the cells.

The reviewer is right and thus we have performed this experiment in cell free PDMS chips. The results are presented in the new suppl. fig. 4. As is indicated indeed when perfusing the basolateral compartment of the chip (while blocking the apical fluid flow) with 100 ng/mL progesterone at a flow rate of 5 μ L/hour, this is indeed partly adsorbed by PDMS in the first 12-24 hours. When the perfusion medium was replaced by a steroid free medium within 12 hours the collected fluids did contain only neglectable P4 levels. From these data we conclude that indeed the PDMS material adsorbs steroids to become saturated with some 24 hours and the adsorbed steroids (likely by hydrophobic interactions with PDMS) were not released when perfused with steroid free medium. This information is included in the text 193-203 and in the new suppl. fig. 4.

Major critics are concerning the in chip fertilisation:

The cleavage and blastocyst formation was significantly lower than in in vitro embryo production.

1) What is the benefit of fertilisation with flow (what is the reason to keep the flow after a confluent layer of BOECs formation)?

The reason to keep the flow, was not because of fertilization, but was because of the epithelial cells, if they have static medium on top of it, they lose their differentiation and polarization, which would make it a not reliable system for embryo production (see also our reply to the two other reviewers).

a. How does this correspond to the physiological flow rate?

We have no specific data about physiological flow rate in cows. We know that by cannulating the oviduct of cows you can collect ampulla and isthmus fluids, with an average volume of 1004ul/24h for ampulla and 500 uL/24h for isthmus, which corresponds to 41 and 20.8 uL/h (Kavanaugh et al., 2002). Whether this is physiologic we cannot say, but by culturing BOECs at 30ul/h, cells were blebbing and did not polarize (see new suppl. fig. 1). That is why we decided to reduce the flow rate to 5ul/h (which is comparable to the lung-on-a-chip model, which also has similar ciliated and secretory epithelium). We included information about preliminary studies on flow rate on lines 130-133.

b. Would a pulsatile flow or an on/off regimen increase the yield?

See answer above: clearly the maintenance of differentiation of the BOEC is strictly dependent on the perfusion speed. We have not tested pulsatile or on/off flow regimes here. We believe the data -we have now- are important and novel. Besides the major point we make on the critical range of apical perfusion speed administered to our BOEC we indeed agree that many additional things (also beyond further variations on the flow speed/ regime) can be designed in the future to optimize the currently developed perfusion system.

c. Why is it not sufficient to perfuse the basolateral side of the chip and avoid any shear stress on the apical side?

Because the epithelial cells need the apical perfusion flow stimulus to become/remain polarized. Although in air-liquid interface this polarity of BOEC can be maintained without perfusion such system is not an option for producing embryos inside the chip. Instead we here successfully designed a system that maintained polarization of BOEC and allowed embryo production.

d. Alternatively, reduce the flow on the apical side?

See point b. indeed one of the many things we and others can work on to further improve this system. Note again that only within a narrow setting of perfusion rates the 5 uL/h was successful. We intend to publish this manuscript in the present form, as it mostly focusses on successful aspects of chip design, and cell culture.

e. Is it necessary to use a chip or would it be sufficient to plate BOECs on a petri dish?

BOECs on a petri dish lost their differentiation and polarization, and their secretion and interaction with gametes are changed (see ref 24). There are studies showing beneficial influence

of co-culture with 2D BOECs. None of them showed insights in creating zygotes in such system nor the comparison of embryos originated from such cultures and in vitro versus in vivo embryos, as we did here. It demonstrates that having a polarized epithelium layer is necessary for this purpose. Information on cell dedifferentiation is included in lines 177-182.

2) The design of the columns is obviously not optimal for the oocytes. The oocytes are trapped and the shear stress is too high. A different chip design would probably increase the cleavage and blastocyst formation. Why did you stick to this design?

When we designed this chip and tested it, we didn't know that the oocytes/embryos would be able to squeeze in and pass it. Before culturing cells in the chips we used agarose particles of different sizes (compatible to oocytes/embryos sizes) to test if the trapping pillars were efficient, and they did trap the particles. So, we used it to validate all the cell cultures and just found out the trapping pillars were not efficient for trapping embryos (because they are more malleable than the agarose particles we used before and can squeeze in between pillars) after all cell culture were successfully validated. At that point, changing the design would also influence our flow rates and made us need to validate the cell culture over again; and we wanted to have this data available to the scientific community soon. Since, we had enough replicates and enough embryos to analyze, we decided to go ahead and emphasized in the text that design must be optimized. We are currently working on a dual chamber design, in which we can have both ampulla and isthmus oviductal cells cultured and connected. It takes a while though to get it designed, successfully fabricate and validated. We include for reviewing purposes proof that the pillars do not allow passing through of particles < 70 μm (see picture below) while mature bovine oocytes and early embryos (with inclusion of the zona pellucida) have a diameter of approx. 150 μm .

Reviewers' comments:

Reviewer #2 (Remarks to the Author):

Most of the reviewer's comments are well addressed. One remaining concern is using the 5mc staining to indicate embryo's DNA methylation establishment. Actually, the second wave of epigenetic reprogramming at this period is far more complex than the overall DNA methylation, in particular of the comprehensive DNA demethylation, difference between maternal and paternal DNA demethylation, as well as the faithfully maintained maternal and paternal imprinted genes. Without investigating the global DNA methylation in embryos, it is difficult to conclude that the oviduct-on-a-chip can significantly improve the epi(genome) during IVP. Actually, methylome experiment is not difficult to conduct and the several other groups have used this method to study the dynamic DNA methylation patterns during preimplantation embryo development. At least, the reviewer recommends that authors should perform the bisulfite sequencing for several key imprinted genes and then compare with the embryos in the control group.

Reviewer #3 (Remarks to the Author):

my concerns have been addressed and the manuscript has been revised satisfactory.

Reviewers' comments:

Reviewer #2 (Remarks to the Author):

Most of the reviewer's comments are well addressed. One remaining concern is using the 5mc staining to indicate embryo's DNA methylation establishment. Actually, the second wave of epigenetic reprogramming at this period is far more complex than the overall DNA methylation, in particular of the comprehensive DNA demethylation, difference between maternal and paternal DNA demethylation, as well as the faithfully maintained maternal and paternal imprinted genes. Without investigating the global DNA methylation in embryos, it is difficult to conclude that the oviduct-on-a-chip can significantly improve the epi(genome) during IVP. Actually, methylome experiment is not difficult to conduct and the several other groups have used this method to study the dynamic DNA methylation patterns during preimplantation embryo development. At least, the reviewer recommends that authors should perform the bisulfite sequencing for several key imprinted genes and then compare with the embryos in the control group.

Thank you for your comment. We do understand that it would be important to measure the DNA methylation by sequencing, and also considering other embryonic stages. But, since the main researcher (Dr. Ferraz) left the lab in Utrecht, we didn't have another student following up the work done in cows for now, hopefully we will have it soon. We have made progress on improving chip design (together with the University of Twente). However, it was not tested with bovine oviductal cells yet. Performing the new experiments requested, regarding the methylation of embryos, will be a bit challenging at the moment and we truly believe that the chip validation and transcriptome data makes it a valuable and novel manuscript for the scientific community. We did change the text toning down our conclusions regarding the epigenetic effects and stating the limitations of the use of immunofluorescence for global DNA methylation quantification.

Reviewer #3 (Remarks to the Author):

my concerns have been addressed and the manuscript has been revised satisfactory.

Thank you.